

# The MetNet vehicle: A lander to deploy environmental stations for local and global investigations of Mars

A.-M. Harri[1], K. Pichkadze[2], L. Zeleny[3], L. Vazquez[5], W. Schmidt[1],
S. Alexashkin[2], O. Korablev[3], H. Guerrero[4], J. Heilimo[1], M. Uspensky[1],
V. Finchenko[2], V. Linkin[3], I. Arruego[4], M. Genzer[1], A. Lipatov[3], J. Polkko[1],
M. Paton[1], H. Savijärvi[8], H. Haukka[1], T. Siili[1], V. Khovanskov[2], B. Ostesko[2],
A. Poroshin[6], M. Michelena-Diaz[4], T. Siikonen[7], M. Palin[7], V. Vorontsov[2],
A. Polyakov[2], F. Valero[5], O. Kemppinen[1], J. Leinonen[1], and P. Romero[5]

[1]Finnish Meteorological Institute, Helsinki, Finland
[2]Lavochkin Association, Moscow, Russia
[3]Russian Space Research Center (IKI), Moscow, Russia
[4]Instituto Nacional de Tecnica Aeroespacial (INTA), Madrid, Spain
[5]Universidad Complutense de Madrid
[6]DAURIA Ltd
[7]FINFLO Ltd
[8]University of Helsinki

*Correspondence to:* Ari-Matti Harri (Ari-Matti.Harri@ fmi.fi)

**Abstract.** Investigations of global and related local phenomenon on Mars such as atmospheric circulation patterns, boundary layer phenomena, water, dust and climatological cycles and investigations of the planetary interior would benefit from simultaneous in situ measurements with a good spatial coverage. Practically, such an observation network would require low mass landers, with a high

packing density, so a large number of landers could be delivered to Mars with the minimum amount of launchers.

The Mars Network Lander (MNL), a small semi-hard lander/penetrator design with a payload mass fraction of approximately 17 % has been developed, tested and prototyped. The MNL features an innovative Entry, Descent and Landing System (EDLS) that is based on inflatable structures. The

EDLS is capable of decelerating the lander from interplanetary transfer trajectories down to a surface impact speed of 50-70 m s$^{-1}$ with a deceleration of $< 500$ g for $< 20$ ms. The total mass of the prototype design is $\approx 24$ kg, with $\approx 4$ kg of mass available for the payload.

The EDLS is designed to orientate the penetrator for a vertical impact. As the payload bay will be embedded in the surface materials, the bay's temperature excursions will be much less than if it was

fully exposed on the Martian surface allowing some savings on mass.

The MNL is well suited for delivering meteorological and atmospheric instruments to the Martian surface. The payload concept also enables the use of other environmental instruments. The small size and low mass of a MNL makes it ideally suited for piggy-backing on larger spacecraft. MNLs



are designed primarily for use as surface networks but could also be used as pathfinders for high-
value landed missions.

## 1  Introduction

Significant progress in several areas of scientific investigation on Mars such as climate circulation,
water-cycle, sedimentary cycle and surface-atmosphere interactions, have been made possible with
spacecraft observations at Mars (Soffen, 1976; Golombek et al., 1999; Smith et al., 2008). In many
investigations significant progress is contingent on good spatial coverage at several locations (Harri
et al., 1999; Harri et al., 2007) with extended temporal and simultaneous coverage, requiring the
concurrent operation of several spacecraft. Current orbital and lander observations are restricted
in spatial measurements primarily due to the low number of active spacecraft available ($\sim 2$) for
making simultaneous coordinated observations.

The payload mass of the launchers, and their cost, restricts the number of spacecraft and instru-
ments that can be delivered to Mars during each launch window. Among the wide variety of science
instruments and payloads relevant to Mars science and exploration some instruments and instrument
types are inherently massive or sensitive, requiring relatively large and massive landing systems to
enable a soft landing. Up to now large landers with multi-disciplinary and complex payloads have
been favoured; Mars Science Laboratory (MSL) is perhaps the ultimate manifestation (Grotzinger
et al., 2012).

The use of lightweight landers would enable the delivery of an observations network to Mars pos-
sibly in a single launch. Meteorology and climate studies are one area of investigation that would
benefit from a network of observations. A lightweight lander would require low mass instruments
with minimal use of resources such as power and heating which is a requirement well suited for
making atmospheric measurements. Heating requirements can be minimised by burying the bulk of
the spacecraft in the regolith and so thermally isolating it from the extremes of the diurnal temper-
ature range on Mars. Burial could be efficiently performed using the inertia of the lander as with
penetrators. Keeping the lander mass low and packing density high would maximise the number of
landers that could be launched towards Mars with a single launcher. This could be enabled by using
inflatable aerodynamic decelerators.

This paper describes the MNL concept, a compact and lightweight vehicle designed to deliver a
set of instruments to the surface of Mars using a combination of lightweight inflatable aerodynamic
decelerators and a penetrator, impacting at a relatively lower, and hence safer, speed compared to
previous high-speed penetrator designs for Mars. Possible uses of the MNL in Mars exploration
along with programmatic and science mission aspects are also discussed. The paper is organised as
follows: history and background of both earlier Mars landers and their Entry, Descent and Landing
System (EDLS) as well as of the MNL design are discussed and described in Section 2; Section 3



provides a more detailed description of the MNL design; precursor missions are outlined in Section 5.1 and potential mission types & scientific applications of the MNL design are outlined and discussed in Section 4. Future prospects are outlined and recommendations made in Section 5.

## 2  Background

### 2.1  Brief overview of Mars lander technologies

The survivability of spacecraft during landing will largely depend on the spacecraft being able to absorb the impact energy without damaging it payload. Landers can be divided into three catagories with the division of these catagories being defined by the landing speed which is an indicator of the kinetic energy required to be dissipated by the spacecraft's landing system.

A soft lander typically touches down on the surface at a speed of around one metre per second using a rocket propulsion system that is initiated at subsonic speeds to control and reduce the speed for a soft touchdown. The advantage of using a propulsion system is that manoeuvres like hazard avoidance, and pinpoint landings are possible. Examples of soft Mars landers are the Viking, Phoenix and MSL (Soffen and Snyder, 1976; Soffen, 1976; Guinn et al., 2008; Grotzinger et al., 2012) landers. Soft landing technology is required for large payloads, heavy payloads and payloads with components sensitive to high mechanical loads.

A hard-lander, such as high-speed penetrators, typically impact the surface at speeds of around $100 \ \mathrm{m \ s^{-1}}$, and experience high decelerations (1000s of gees) over short time periods during the penetration of the subsurface strata. The use of high-speed penetrators for planetary science were first studied in the USA during the 1970s. The Soviet Union seems to have initiated its studies in the 1980s (Ball et al., 2009, Chapter 19). Penetrators for a variety of Solar System destinations have progressed to the concept stage although only two designs have actually been launched. These are the Russian Mars-96 penetrator (Surkov and Kremnev, 1998) and USA's Deep Space 2 Mars Microprobe (Smrekar et al., 1999). Each missions included two penetrators riding piggy back on a carrier spacecraft. None of these penetrators were successful: the Mars-96 mission failed to reach a stable Earth orbit and the DS-2 probes' fate after deployment from the Mars Polar Lander is not known. Hard landers provide a platform to take robust science payloads to a planetary surface with a high mass efficiency. This is because the more gently a vehicle lands the more mass is needed for the EDLS to decelerate the vehicle's velocity before the touchdown on the surface.

Semi-hard landers are vehicles that impact the surface at speeds, and experience subsequent decelerations, that are between those of a soft lander and a hard lander. Typical Martian semi-hard landers are, e.g., the Mars Netlander vehicle (Harri et al., 1999) and the Mars-96 Small Station (Linkin et al., 1998) both of which were using the heat shield, parachutes and airbags in the entry, descent and landing phase. Semi-hard landing vehicles impact on the surface with a moderate deceleration (few hundreds of gees over the time of some tens of milliseconds) and thus provide a



practical solution for a planetary surface payload including robust geophysical instruments. That
is especially suitable for instrumentation including lightweight sensor systems needed to perform
atmospheric science experiments.

## 2.2 MetNet Lander development history and background

The work on a semi-hard lander design for the MNL started in August 2000. Five different EDLS
concepts (Table 1 and Fig. 1) were initially defined as candidates to be studied. The development
of the MNL design was performed over a 7 year period from 2001–2008 by a team comprising
of FMI, the LA and the Russian Space Research Institute IKI. The Spanish INTA joined the team
in 2008. The MNL development work was funded and led by FMI. The MNL concept and key
probe technologies were developed and the critical subsystems were qualified to meet the Martian
environmental and functional conditions during the years 2002–2005. Development of the required
system instrumentation and prototype science payloads to facilitate testing was carried out in 2004–
2008.

In the initial phase of the development five different EDLS concepts were assessed from the view-
point of finding an optimal solution for deployment of small payloads onto the Martian surface. One
concept was a traditional, parachute-based and the remaining four utilized inflatable structures in
various ways.

**Table 1.** The studied MNL EDLS concept candidates. In each concept the entry and descent phase braking
devices are jettisoned to reduce decelerated mass. The concept in the first row is as Mars-96 Small Stations
(Linkin et al., 1998). The column title 'Entry' refers to the hypersonic and supersonic portion of the flight.
'Descent' refers to the subsonic portion of the flight. A 'tension cone' refers to a type of inflatable decelerator
shaped so as to contain tensile stresses, e.g. see Clark et al. (2009) for more information.

| Concept | Entry | Descent | Landing | Station type |
| --- | --- | --- | --- | --- |
| A1 | rigid shell | parachute | airbags | lander |
| A2 | rigid shell | tension cone | airbags | lander |
| B1 | rigid shell | tension cone | internal shock absorber | penetrator |
| B2 | rigid shell | inflatable torus | same as descent | lander |
| B3 | inflatable | attached ballute | internal shock absorber | penetrator |
| selected | inflatable | tension cone | internal shock absorber | penetrator |

Comparative analysis between the five concepts, underlining and emphasising reliability, payload
fraction and complexity of test programme, was carried out. The concepts were catagorised into two
catagories. Catagory A contained those landers using airbags for landing and catagory B contained
those landers using other impact shock attenuation mechanisms for landing. These catagories con-
tained a range of variants as shown in Fig. 1 whose EDLS elements are listed in table 1. Variants
A1 and B3 were selected for additional, more detailed study. This study resulted in the formation of



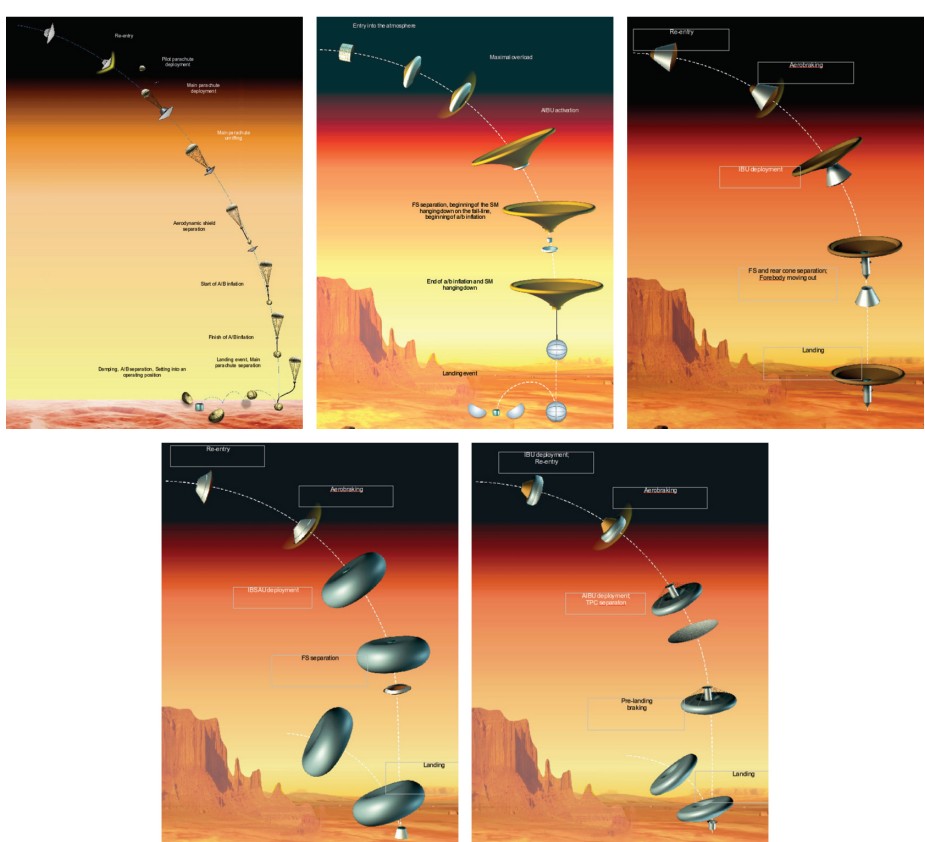

**Fig. 1.** Landing schemes and designs (from left to right and top-down: A1, A2, B1, B2, B3 as in Table 1) investigated during the course of development of the MNL concept.

lander concepts known as concept A and concept B. Concept A was essential variant A1, based on the Mars 96 small station, which employed a rigid heat shield, parachutes and airbags. Concept B was a new formulation of the EDLS that employed an inflatable heat shield, tension cone and pen-

115  etrator to deliver the lander to the surface. Concept B was found to have the best overall reliability because the EDLS used a lower number of pyrotechnics devices and control commands than concept A. A high payload mass to entry mass ratio is also possible with concept B because penetration into the Martian regolith dampens the diurnal temperature variations and reduces the thermal control requirements. The current MNL design was chosen as it proved to best satisfy the design goals and

120  criteria.



### 2.3 Selected Entry, Descent and Landing System concept

The selected MNL Entry, Descent and Landing System (EDLS) was designed to cope with entry speeds of slightly over $6\,\mathrm{km/s}$. The major components of the EDLS are the Hypersonic Inflatable Braking Unit (H-IBU), the Transonic Inflatable Braking Unit (T-IBU) and penetrator. The H-IBU is an inflatable heat shield designed to resist the heat during hypersonic entry into the atmosphere and decelerate the vehicle down to supersonic speeds. The T-IBU is an inflatable device known as a tension cone and is designed to decelerate the vehicle from supersonic speeds, through the transonic region down to subsonic speeds. Soon after the T-IBU is deployed the H-IBU is jettisoned. Once the H-IBU is jettisoned the forebody of the penetrator is deployed and locked into place ready for impact with the surface

A MNL can be separated from the carrier spacecraft either directly from a Mars-approaching trajectory or from Martian orbit. Depending on the mission concept, a single carrier spacecraft may carry and deploy a single or several MNL. During the Earth-Mars cruise and possible orbital injection the carrier spacecraft provides each MNL with communications (data link) and power (for instance for health checks every few months, software upgrades, etc.) through the Carrier Spacecraft Interface and Lander Deployment System (CSI-LDS). The CSI-LDS features may vary depending on the number of MNLs carried, the mission concept and the characteristics of the carrier spacecraft.

A proposed MetNet mission with 16 landers was made in 2007 as a study for a European Space Agency (ESA) medium class mission. Each lander was allocated a mass of $20\,\mathrm{kg}$ plus $10\,\mathrm{kg}$ for the spin/ejection mechanisms. The estimates were given with a margin of 10-20%. The Entry, Descent and Landing (EDL) sequence of activities, shown in Fig. 2, begins with the *separation* phase from a few hours to a few days before actual separation from the carrier spacecraft. The MNL batteries (Section 3.2) are charged to capacity and depending on what has been performed during the preceding health check, final parameter updates to the Command and Data Management System (CDMS; *e.g.*, software, cyclograms – see Sections 3.3 and 3.5) may also be made. Just prior to separation the MNL clock is set and the lander is spun up for stability during the entry into the atmosphere. This process takes $< 10\,\mathrm{min}$ to complete.

Since the MNL itself does not have thrusters for trajectory or attitude changes, the carrier spacecraft may also need to carry out attitude change manoeuvers to eject each MNL at the correct angle and at the correct time to reach its intended landing area (the eventual landing site is also influenced by atmospheric parameters during the EDL) and return to the desired attitude after the separation.

The behaviour of an MNL during the entire EDL is monitored by a combined 3-axis accelerometer and gyroscope instrument. This diagnostic information is transmitted in packets in near-real time to the relay spacecraft (the carrier or a Mars orbiter, if one is suitably positioned during the EDL) via two dedicated beacon antennas (Section 3.3). The CDMS of the MNL connects the radio system first to the outer beacon antenna, after the inflation of heat shield to a second antenna and after

landing and deployment of the instrument mast to the main antenna. The data packets include lander identifiers, hence the monitoring system permits overlap or concurrence of EDL phases of multiple landers.

The *entry* phase begins when the MNL senses the first indications of interaction with the atmosphere and ends in the transonic (transition from super- to subsonic) speed regime. The inflatable heat shield is deployed during the entry phase to stabilise and decelerate the lander. The optimal range for the entry angle is -16 to +18 $\pm$ 2°. The inflatable heat shield diameter is 1 m which decelerates the vehicle down to a Mach number of about 0.85 at an altitude of 4.5-11.0 km and dynamic pressure of 95-130 Nm$^{-2}$ (both altitude and dynamic pressure depending on the angle of entry). The tension cone is fully inflated and the heat shield released 10 s later allowing the vehicle to stabilise.

The *descent* phase begins when the lander speed is below the transonic regime, the inflatable heat shield is ejected and the tension cone is deployed. The tension cone diameter is 2 m, and is used to decelerate the MNL down to a landing speed of 47-55 m/s, depending on the angle of entry, at the Martian datum, i.e. the point of zero elevation on Mars equivalent to the altitude where the pressure is 610 Pa. The descent phase ends with the contact of the penetrator tip with the surface. Peak deceleration of the MNL payload bay during the impact will be <500 g and the total impact time is 20 ms. The minimum impact speed required for an operational landing is 50 m/s with a maximum horizontal wind speed of 20 m/s.

The *landing* phase begins when the tip of the penetrator touches the surface and ends when the lander has come to rest on and is partially embedded in the top layers of the surface. The deceleration experienced by the payload as the lander penetrates the surface is of the order of 500 g. The structures and mechanisms involved in the final phase landing process, comprising of the Shock Absorbing System (SAS) which is used to reduce the g-levels on the instruments, are described in greater detail in Section 3.1.

## 3   Description, operation and testing of the prototype hardware

### 3.1   Structures and mechanisms

The MNL mechanisms are divided into two categories which are a) the Entry, Descent and related subsystems and b) the Landing and surface operation related subsystems. The Entry and Descent System consists of three subsystems which are 1. Rigid Aerodynamic Shielding (RAS) and supporting structure, 2. Flexible Heat Protection (FHP) and 3. H-IBU (see section 2.3), inflation system and load-bearing elements. The landing and surface operation system consists of three subsystems which are 1. T-IBU and gas generator, 2. Surface Module (SM) with a Shock-Absorbing System (SAS) and 3. Equipment Compartment (EC).

During cruise, entry and most of the descent phase, the EDLS related subsystems and the SM are efficiently packed in terms of volume. This is achieved by stowing the systems telescopically



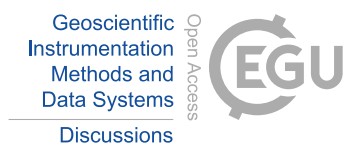

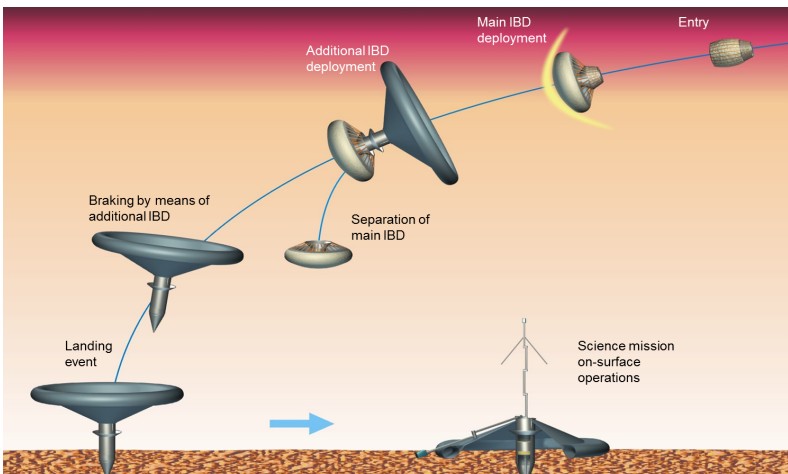

**Fig. 2.** MetNet Lander (MNL) Entry and landing sequence and configurations at different parts of the sequence: 1) The complete lander in stowed configuration during cruise and coast phase prior to atmospheric entry; 2) Main Inflatable Breaking Unit (MIBU) deployed for atmospheric entry; 3) Additional Inflatable Breaking Unit (AIBU) deployed, MIBU and Rigid Aerodynamic Shielding (RAS) not yet jettisoned; 4) Surface Module (SM) in landing configuration with the forebody deployed.

inside each others where possible. The forebody is stowed inside the surface module cylindrical structure using the empty volume, once deployed for the deceleration of the equipment compartment during the impact with the surface. The forebody will be deployed into a landing configuration after jettisoning the H-IBU. The stowed surface module and forebody are both stowed inside the mechanical support cylindrical structure of the rigid section of the front shield during entry and upper atmosphere braking phase. Fig. 3 shows the complete lander with empty stowed H-IBU wrapped around it.

The SM accommodates the system electronics and payload instruments. The T-IBU is connected to and surrounds the surface module. These three subsystems stay interconnected after touchdown forming the surface operating unity. The power system solar cells are attached to the upper surface of the T-IBU. Other subsystems, forming most of the EDLS, are ejected during the descent phase as shown in Fig. 2. The SM accommodates the EC, which houses the system and payload electronics and supports external sensors (the boom with meteorological sensors, optical sensor) and telecoms antenna. The SM includes a rear cover lid, which protects the module during entry and landing.

The SM includes the shock-absorbing system (SAS). This system allows the Equipment Module (EM) to slide some tens of centimetres during the impact with the surface and thus reduce the deceleration experienced by the equipment module by a factor of around two compared to other rigid mechanics such as the surface module body structures. The SAS is made of six metallic (AMg3M



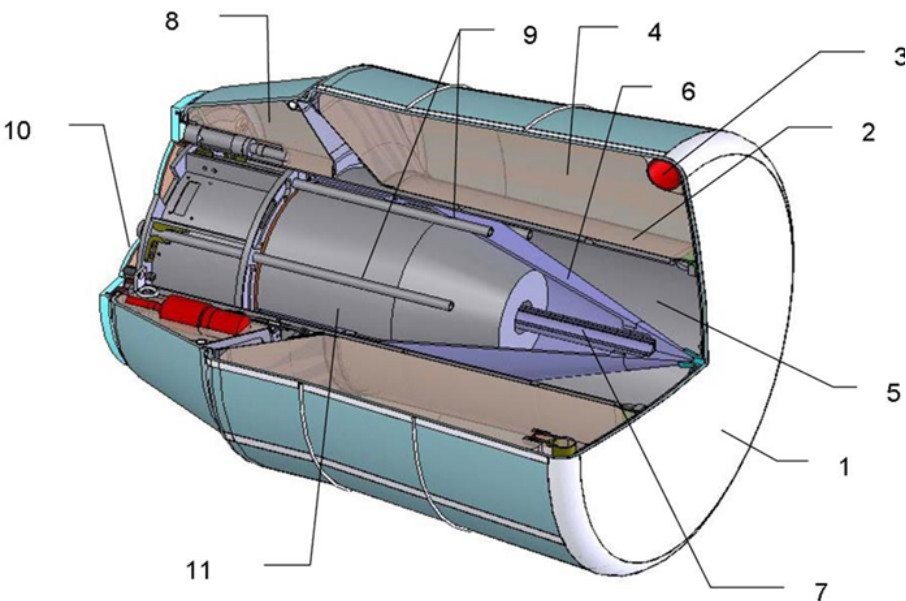

**Fig. 3.** The Rigid Aerodynamic Shielding (RAS) includes a blunt front shield plate, a toroidal pressure vessel (which stores the H-IBU inflating gas under pressure) as well as supporting structures for both the Surface Module and the entry and descent systems.

Aluminium-Magnesium alloy) hollow tubes. The equipment module slides along these tubes during the impact on the surface and kinetic energy is reduced by squeezing the hollow tubes flat by squared sliding slots of the equipment module supporting adapter.

### 3.1.1 Entry and descent related subsystems

The RAS including structural details is shown in Fig. 4. The surface module with stowed forebody fits inside the cylindrical and conical structure. Fig. 4 (c) shows the surface module with the forebody extended. The surface module and RAS are connected and secured together by a cable and two turn-buckle devices. The RAS blunt circular front shield has radius of curvature of 1.0 m and a diameter of 0.46 m. The RAS is manufactured from AMg6, AMg6M and MA2-1 aluminium-magnesium

alloys. The H-IBU interfaces with the RAS by a H-IBU-Inflation System ( H-IBU-IS). H-IBU-IS includes a toroidal pressure vessel for storing the H-IBU inflation gas and required pyro operated valves. The toroidal H-IBU-IS can be seen in Fig. 5 surrounding the circular front shield. The RAS has total mass of 2.31 kg.

The H-IBU consists of a toroidal inflatable wheel, shown in Fig. 5 (b) which supports the flexible

TPS, increases the frontal braking area and maintains the stability and flight path angle during early



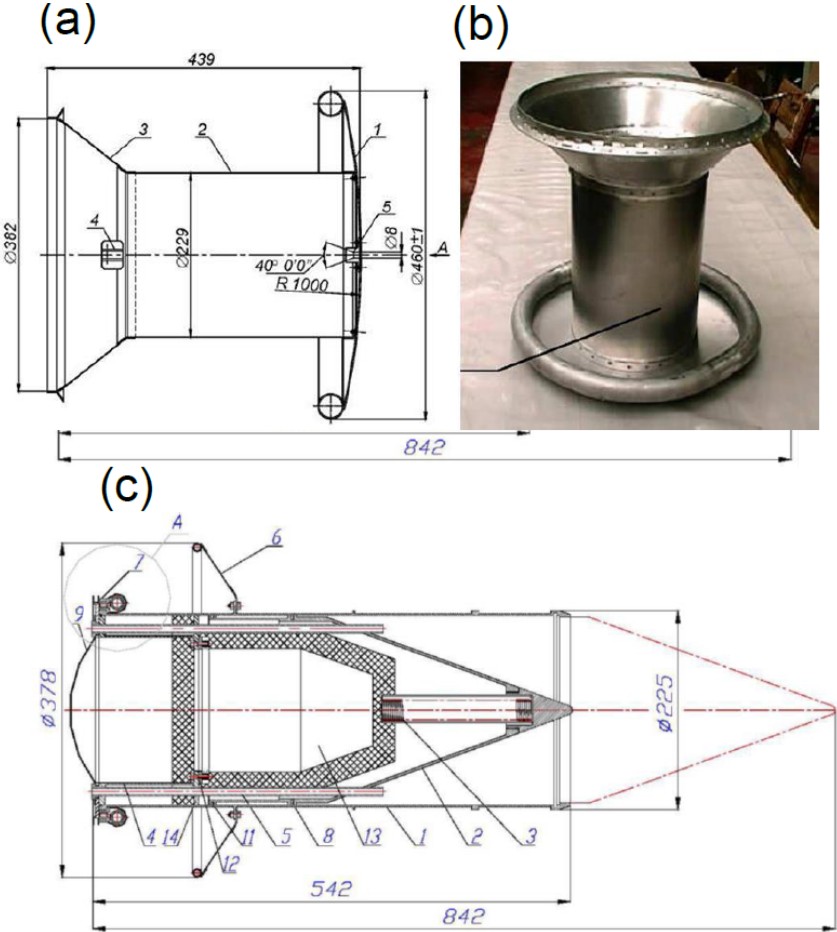

**Fig. 4.** The top two figures show the Rigid Aerodynamic Shielding (RAS) which includes blunt front shield plate, toroidal pressure vessel which stores the H-IBU inflating gas (under pressure) and supporting structures for the Surface Module and entry and descent systems. Image (c) shows the SM in stowed configuration (1) main body (2) conical forebody (3) spring to deploy forebody (4) IB (5) deforming tubes (6) conical structure (9) lid for protecting external and deployable instrumentation (13) EM.

landing phase within specifications. The H-IBU consists of 12 tubular segments each having a diameter of 250 mm. The total diameter of the complete inflated H-IBU is 1000 mm. Inflation pressure is 63 Pa. The H-IBU consists of an internal gas tight bladder (TPM-8 fabric), external cover fabric (aramid fibre), load bearing tapes, filling hoses as well as hardware and accommodation
bag. Total mass is 1.17 kg. Fig. 5 (c) shows the inflated H-IBU.



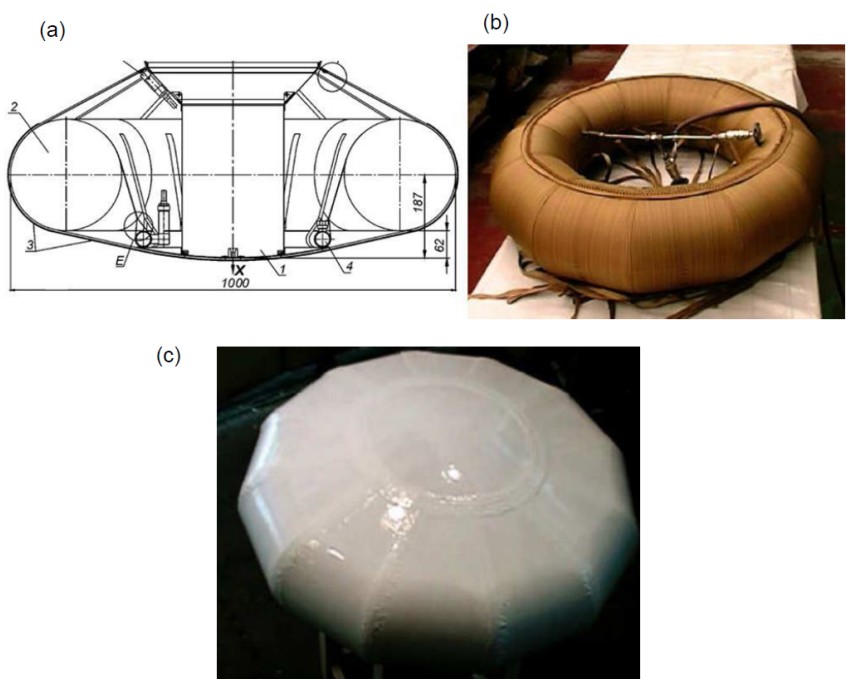

**Fig. 5.** Images (a) and (b): Main Inflatable Braking unit ( H-IBU) in the drawing on the left is a toroidal inflatable wheel (4), which supports the flexible heat protection system (3) and maintains required shape for maintaining correct attitude and flight path angle. Torus shaped pressure vessel/pressure receiver (4) provides required inflation gas. Inflated H-IBU assembly with the cover shell is shown on the right. The bottom figure shows the Thermal Protection Shell (TPS) deployed. It is supported by the rigid section of frontal shield in the middle and toroidal inflatable Main Inflatable Braking Unit ( H-IBU) constructed from 12 segments.

### 3.1.2  Landing and surface operation related subsystems

The T-IBU, shown in Fig. 6, is used during the last stages of the descent and landing. The T-IBU is deployed just before jettison of the combined H-IBU and RAS and supporting structures. The T-IBU deccelerates the MNL down to subsonic speed and stabilizes the lander. After landing T-IBU also
supports the solar panels, which are mounted on its surface.

The T-IBU consists of toroidal shell cover and gas tight bladder, flexible cone and inflation system which is based on a pyrotechnic gas generator. The toroidal part of the T-IBU is made up of 12 segments with each segment having a diameter of 200 mm. The inflated T-IBU has an overall diameter of 1800 mm. The T-IBU hardware is accommodated into a cone shaped upper part of the
External Body of the SM. The T-IBU has a mass of 1.06 kg.

The SM is the final stage. The surface module consists of the Equipment Module (EM), Inter-




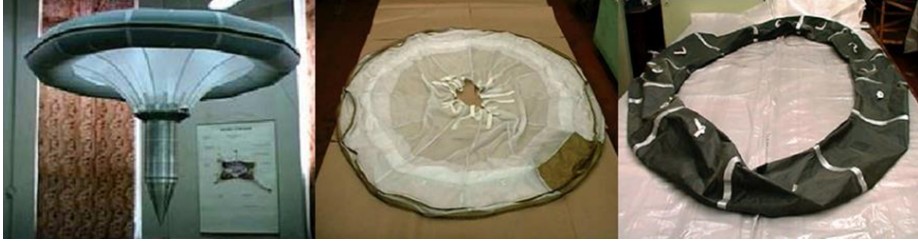

**Fig. 6.** Inflated Additional Inflatable Breaking Unit ( T-IBU) shown on the left with the Surface Module (SM). In the center the cover of the T-IBU and on the right the gas tight bladder.

nal Body (IB), the Forebody (FB) and the Main Body (MB). They are made of AMg6 aluminium magnesium alloy.

The forebody is conical and is a telescopically extending ground penetrating forward section of
the surface module. During the last stage of deployment the forebody is locked together with the MB thus forming a unified structure. The IB is a cylindrical compartment inside the upper section of the MB. The IB is mounted together with the EM below it. The IB accommodates the external deployable instrument boom. The EM is sealed and thermally insulated from the environment and accommodates electronics.

The SAS (Shock Absorbing System) is installed inside the MB, Fig. 7 (a), and is designed to reduce g-loads on the payload compartment. The SAS is based on six deforming hollow tubes, that can be seen in Fig. 7 (b), mounted inside the MB. These tubes support Internal Body and EM, which during landing impact slide along these tubes. The IB has narrower guiding slots for the tubes than their external diameter. The tubes will thus be deformed narrower accordingly and absorb the
kinetic energy from the EM during the impact with the surface. The IB and EM can decelerate over a distance that is 30 cm more than the combined MB and forebody. Using this method the deceleration can be limited to a maximum of 500 g. The tubes are made of aluminium magnesium alloy AMg3M.

Fig. 8 shows the configuration of the MNL's internal and external components just before and after landing. In Fig. 8 (b) the payload compartment (EM and IB) has slided downwards along the
six deforming hollow tubes.

The payload compartment consists of the EM and IB mounted together. The EM is sealed inside thermal insulation and accommodates most of the payload electronics and batteries. On the top of the EM is the IB, which accommodates the instrument boom. The boom supports temperature and humidity sensors as well as the camera and optical sensor. The boom also supports the telecoms
antenna. The IB features the interface with the SAS as described earlier. Fig. 8 (c) shows the payload compartment.



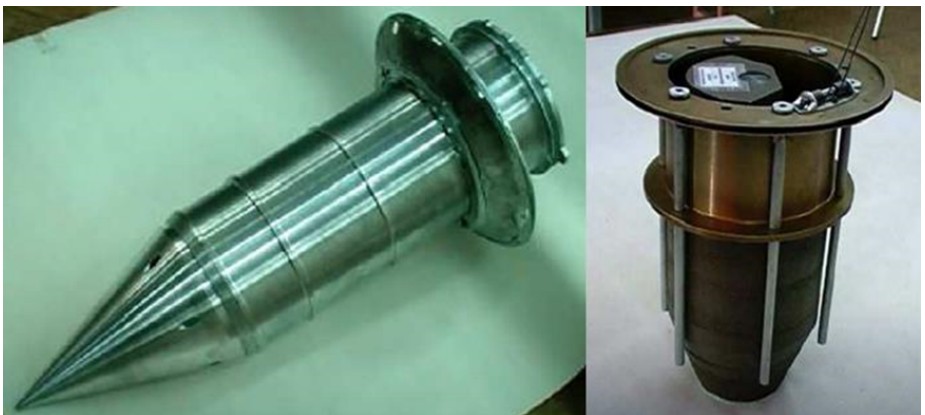

**Fig. 7.** On the left is shown the Surface Module forebody deployed. On the right there is the Internal Body together with Equipment Module mounted below it. Shock Absorbing System tubes are visible. The hollow tubes absorb kinetic energy by deforming as they slide through too small openings on the flanges of the Internal Body.

### 3.2 Electric power and thermal management subsystems

The primary power source for the MNL baseline design is solar energy. Flexible Si solar cells with cell dimensions 11.4 cm × 4.6 cm and total area approximately 400 cm$^2$ are placed in pockets
sewn on the fabric of the upper side of the T-IBU. The cells provide daily average electrical power of about 600 mW. Energy storage of about 40 Wh is provided by two SAFT MPS176065 Li-ion batteries connected in series inside a thermally sealed container. Originally inclusion of RTGs into the basic design was investigated and consequently the design does accommodate them. The RTGs were dropped, however, due to anticipated precursor mission options, due to difficulties related to
availability of the devices and due to environmental impact assessment, political and security issues related to launching radioactive materials and devices.

     This baseline non-RTG power system design limits operations to latitudes effectively between ±30° about the equator (see also Fig. 16) and even within that latitude band night-time operations will be highly constrained by available power. During times of increased opacity of the atmosphere
due to dust storms, during local winter time at higher latitudes or in case parts of the solar cells are covered by dust, the generated electrical energy will be reduced, limiting the operational possibilities further. Inclusion of a Radioisotope Thermal Generator (RTG) would allow for more continuous and robust operations as well as landing sites and operations also during wintertime at higher latitudes, up to polar regions.

The rather limited amount of power (ultimately dissipated as heat necessary for thermal management) provided by the non-RTG power system is partially compensated by the passive thermal





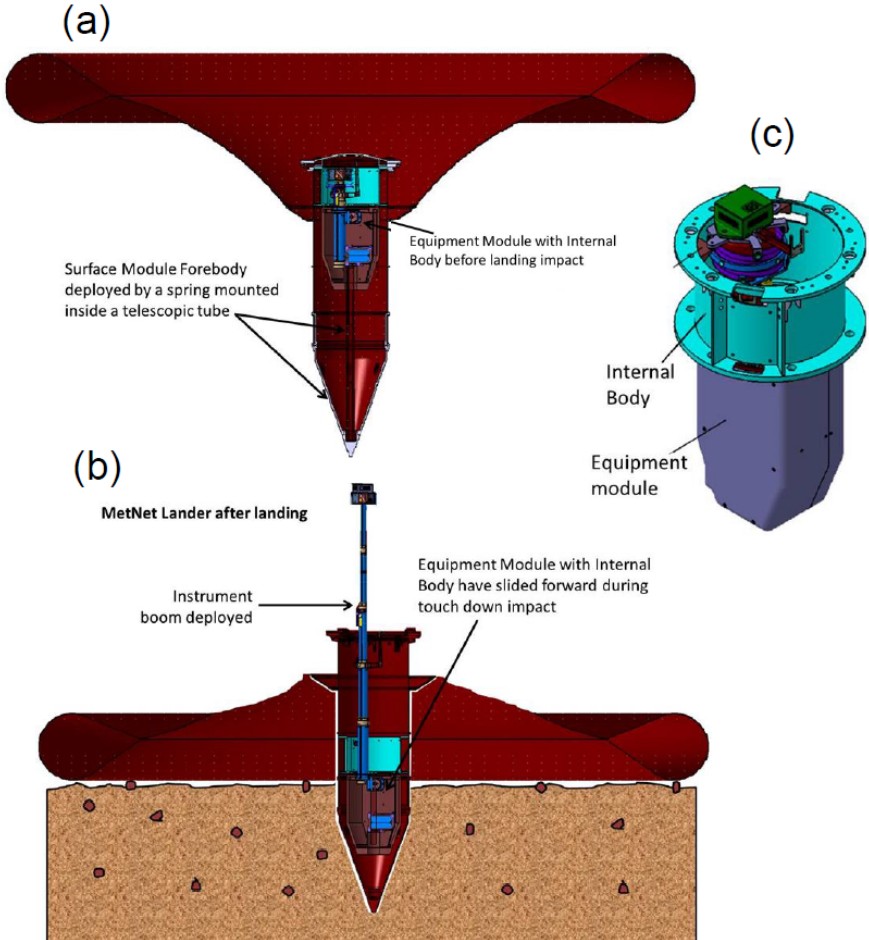

**Fig. 8.** Image (a) shows the configuration of the external and internal components of the MNL in the last stage of the descent before impacting the surface. Image (b) shows the configuration of the same components after impacting the surface. Image (c) shows a perspective view of the internal body and equipment module.

control inherent in the penetrating MNL design: after a successful landing the front part is submerged in the Martian soil and in good thermal contact with its surroundings. Since the amplitude of the temperature variations tends to decline fairly rapidly with increased depth (Fig. 9), this results in

smaller thermal variations for those payload components and lander subsystems housed in the front part of the lander. The MNL battery can operate down to temperatures of 220 K and will have its own additional thermal insulation and heaters to increase the battery temperature during charging to at least 250 K if needed. The parts and subsystems remaining above the surface face comparatively much harsher thermal environment.




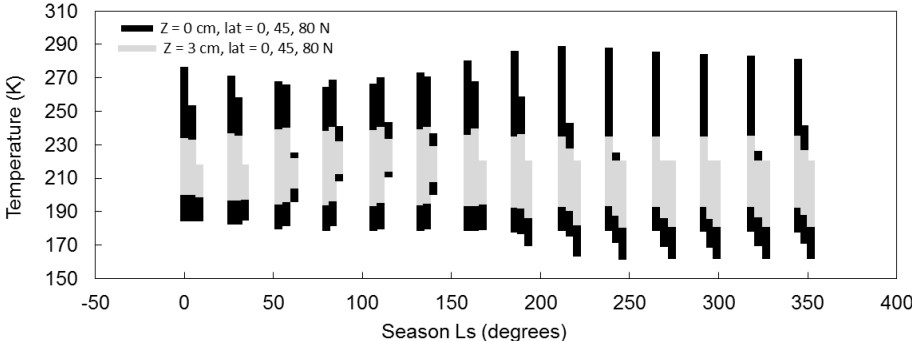

**Fig. 9.** The range of temperatures experienced at different latitudes and depths on Mars over the Martian season. The light grey bars represent the surface temperature and the black bars represent the temperature at a depth of 3 cm. The bars a grouped together in bunches of three representing the three different latitudes modellied, 0, 40 and 80° N.

## 3.3 Communications, CDMS and electronics

The MNL does not include direct MNL-Earth communications capability – a relay spacecraft [either the carrier spacecraft during Earth-Mars transit or a Mars orbiter] is required. Observational and housekeeping data are preprocessed (including image compression) on board the MNL and transmitted to a relay orbiter on the UHF band. The radio system is built around the same type of micro controller used in the CDMS. Together with an FPGA it implements the Proximity 1 protocol (REFERENCE) (for compatibility with the current and likely future Mars orbiting platforms). The system supports a bi-directional data link while still connected to the carrier vehicle, allowing a full system checkout as well as last-minute adjustments of operational parameters. The communication system is also capable of supporting a bi-directional link. There is also a technical capability to support software updates.

The CDMS is built around a fully redundant micro controller system where one system is capable of autonomously detecting and correcting errors in the performance of the active controller. All hardware interfaces and memories are duplicated so that the secondary controller can operate the system completely in case the primary one malfunctions without correction possibility. A block diagram is shown in Fig. 10. The software and even the controller hardware configuration can be updated from the operational controller via the implemented JTAG (DEFINE) input, using the own configuration as reference. The monitoring between the redundant controllers is done by a bi-directional CAN-bus interface integrated into each of the controllers. This link is also used to update cyclogram contents in the secondary controller after a commanded update.

Each of the MNL sensors are required to pre-process its observational data including image compression inside the panorama camera. The controller itself does not include any general data com-



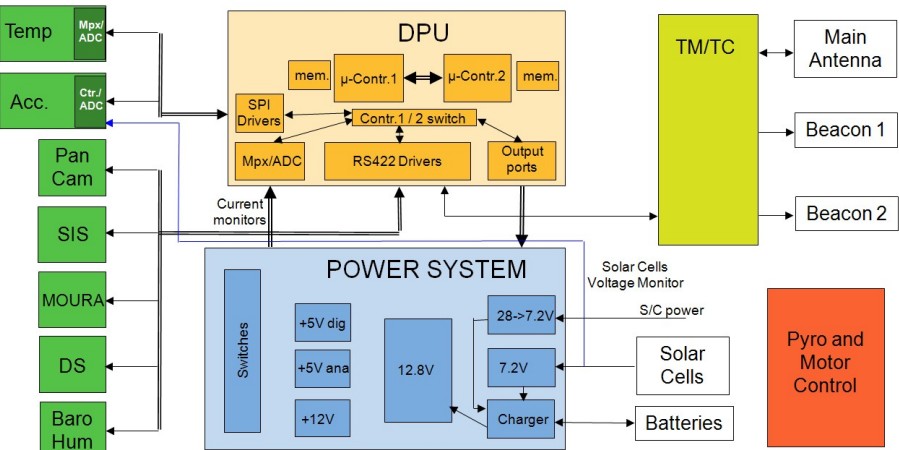

**Fig. 10.** The MNL electronics consists of a hot-redundant microcontroller with duplicated interfaces, a power conditioning and a radio system. Communication with the detectors is via serial links.

pression software to minimize especially energy resources.

### 3.4 Payload resources and strawman payload

With a total of 4 kg for the payload including 2.6 kg for the control system and meteorological
mast including antennas, 1.4 kg are available for the instruments. As instruments are normally not
operated in parallel the maximum available current from the batteries has only to be shared between
the CPU and one sensor. With a maximum current of 5.8 A at 2 * 3.5V about 40 W maximum power
is available for a short time (of the order of 1 h; see also Section 3.2). The average power used has to
be balanced against the average power provided by the solar cells and limits the time an instrument
can be operated per sol. Instruments and their electronics can be accommodated either inside the
thermally stabilized payload compartment guaranteeing temperatures above -50 C or outside on or
close to the telescope mast. The payload compartment is illustrated in Fig. 11.

The strawman payload for the precursor mission includes sensors for temperature, pressure and
humidity measurements, a 4-lens panoramic camera, a multi-band spectrometer with 2-pi view, a
3-axis magnetometer, a dust sensor and a combined 3-axis accelerometer / 3-axis gyrometer for
descent control and monitoring.

### 3.5 MNL operations

Since commanding of a MNL is probably possible only infrequently if at all, a highly autonomous
operations concept is necessary. The driving design factors constraining the vehicle design are op-



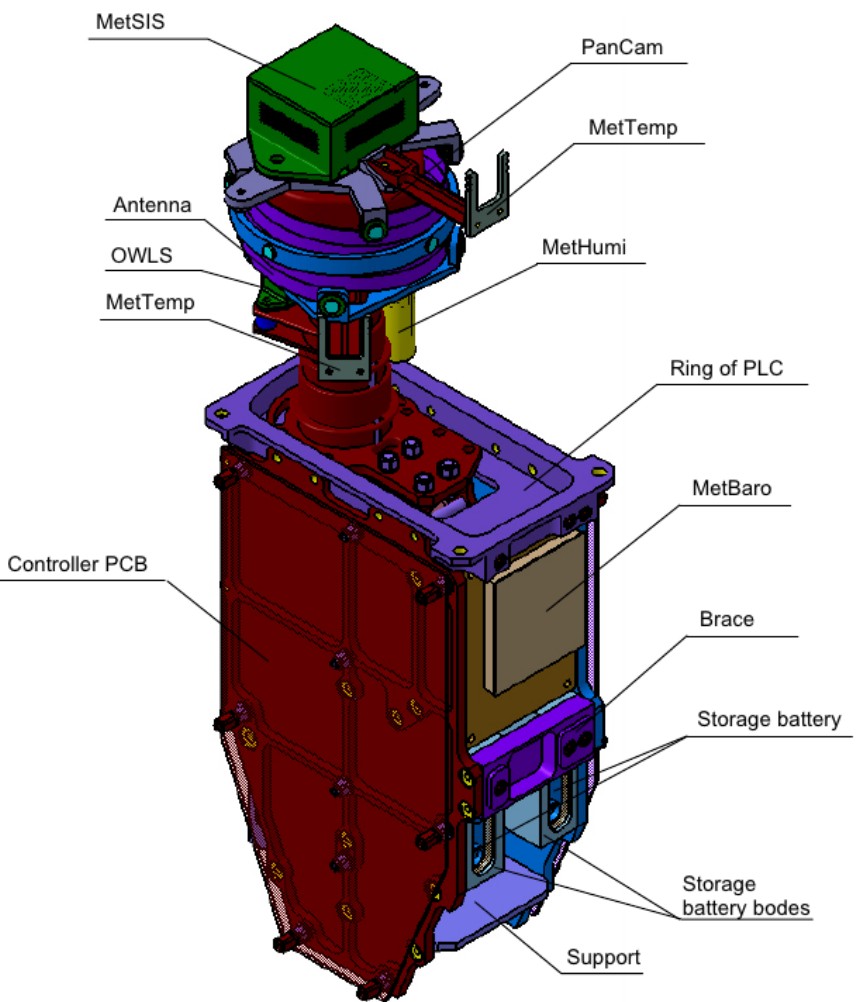

**Fig. 11.** The payload compartment of the MetNet Lander slides after impact some 30 cm with deforming struts absorbing the kinetic energy remaining at the time of impact. All the instruments are packed tightly together with the CDMS and other system electronics.

timal utilisation of the limited energy and availability of telemetry link time. The system also has to be able to adapt to different environmental conditions (*e.g.*, day/night) and correct or minimise impacts of system problems. The general control scheme is illustrated in Fig. 12.

  Due to the limited power supply, telemetry sessions exclude simultaneous observations. Phobos eclipse observations (see end of this section) will also take precedence over regular observations.





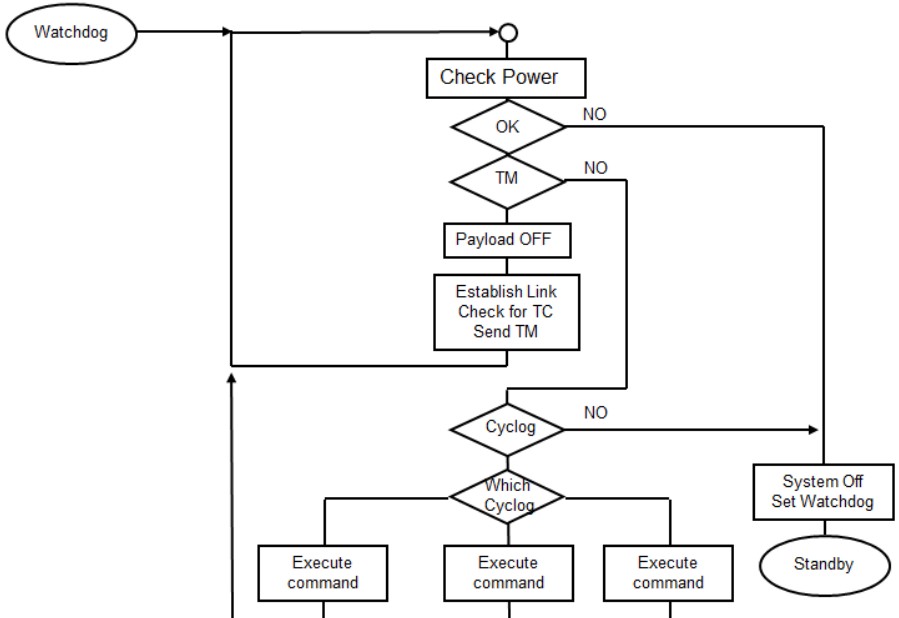

**Fig. 12.** The control system consists of an infinite loop with first a sequence of resource checks, followed by
the execution of the next cyclogram command if possible. In between activities the controller is in low-power
wait state.

The system software is implemented as a linear process, started at regular intervals of about $20\,\mathrm{ms}$
out of a low-power stand-by mode, checking first the battery status, then the availability of a teleme-
try link. If either the energy level is dangerously low or a telemetry link is possible, any science
operation is aborted and all sensors are powered off. Telemetry remains switched on as long as
relay link energy and non-transferred data are available. At other times observations are carried
out according to a *cyclograms* concept. As up to 20 cyclograms – specifying detailed timing and
command sequences for the sensors – can be defined, a sufficient set of scenarios can be covered.
Cyclograms can be updated, if commanding is available. The concept allows consolidation and
freezing of the system software at an early stage for testing while the detailed operational sequences
may be optimized at least up to close to lander separation.

Concerning the electrical energy, battery charging from the carrier spacecraft or from the solar
cells is completely controlled by hardware. Information flow between the controller and sensors or
the telemetry system is handled by an independent processor inside the controller. This processor
handles real-time tasks like sending of commands or reception of data packets, their consistency
checks and acknowledgements. As conversion between internal parallel data words and external



serial bit streams is also handled by autonomous hardware inside the controller, the processor is most of the time in stand-by mode, reducing the energy needs.

Each cyclogram is stored as a matrix with a control header, followed by many fixed-length command vectors. The header contains conditions under which the cyclogram may be used and a pointer to the next command vector to be executed. Additionally each vector defines the time delay to the

following vector and a set of conditions under which this vector has to be skipped. This way the same cyclogram can be used under different conditions like day or night, solar incident angles dangerous for an optical sensor or in case one of the addressed sensors is defective. The command itself contains the address of the system part or sensor, the command code itself and possibly a number of parameters associated with the command. In case the parameter list is too long to fit into the

short vector space, the list is replaced by a pointer to a list at the end of the main matrix. To simplify writing and verification of the cyclogram details, the most common command combinations are hard-coded as macros and can be called directly from inside the cyclogram like a direct sensor command, shortening the cyclogram significantly.

The adaptivity of the system is based on the condition flags used to select a given cyclogram

and to possibly skip a command vector inside the active cyclogram. The cyclogram control system interprets general system conditions and some of the detector measurement data to define the status of these flags. As the different cyclograms contain possible combinations of sensor operations, mainly sequential, but possibly also simultaneous, an additional adaptive priority scheme ensures that each sensor gets a just share of the observation time suitable for its operation. A change in

environmental or system conditions during the execution of the main control loop may result in the currently active cyclogram being aborted: the related sensors are switched off and the next possible cyclogram started. This ensures a maximum scientific return with the available resource limitations.

Criteria based on sensor measurements can be refined by programmable algorithms defining the upper and lower limit of these parameters before a condition change is initiated. The most important

selection criteria are listed in Table 2 and additional criteria can easily be implemented.

A special case is observation of Phobos eclipses, used to locate the actual landing site with high precision. This in turn allows for correlation of the time stamps of the data with the correct local time, thereby making comparisons between observations from different landers more precise.

A time window for the passing of the Phobos-moon's shadow and for the high-time-resolution

measurement with the optical sensor at the top of the mast needs to be determined and pre-programmed into the MNL control system. This can be calculated once the landing area is known based on the trajectory of the transfer vehicle and the scheduled moment of release of the lander. Only daytime passes are observed and only one diode will be used to keep the data amount as low as possible. This eclipse-mode is controlled directly by the system software *outside* the cyclogram scheme. When the

programmed absolute time is reached, the active cyclogram is aborted and the macro-like eclipse program with a finite loop is started. The data stored overwrite the not yet transferred data from



**Table 2.** Environmental or system status cyclogram selection criteria.

| Criterion | Determination |
|---|---|
| Low battery status | Information comes directly from the power subsystem. Can be used to activate only sensors with low energy needs (*e.g.*, T, p, H), when the battery charge is low, but not yet critical. |
| Critical battery status | Information comes directly from the power subsystem. |
| Data buffer availability | The telemetry handling system keeps track of the data buffer usage per sensor. If the usage limit is reached, the sensor will be disabled and related commands skipped in active cyclograms until data has been uplinked and space made available. This limitation can be ignored to allow collection of large amounts of data under special conditions. |
| Severe detector error | To save energy and time resources, a detector is disabled, if it repeatedly fails to react on commands appropriately. Recovery will be tried once per sol or after a system failure with subsequent reboot. |
| Day/night status | Determined from data from the optical sensors at the top of the mast. During the night the camera and the optical sensor will be disabled. |
| Solar incident angle | Based on accelerometer data the solar incident angle to the infrared dust sensor is calculated disabling its operation when directly illuminated. |
| Humidity | Humidity sensor data allows for selection or de-sele ction of the optical sensor $H_2O$ spectral band. |

other sensors if necessary. The measurement is repeated three times about 7.5 h apart to cover the theoretically expected three subsequent eclipses.

### 3.6 Aerodynamic and Aerothermodynamic considerations

The aerodynamic and aerothermodynamic properties important for the design of the EDLS and trajectory have been studied in detail for MNL using laboratory tests and numerical modelling. Both the MNL with the heat shield inflated and the MNL with the tension cone deployed have been studied with wind tunnel tests and computer models to determine their aerodynamic, dynamic stability and static stability coefficients. Structural analysis of the penetrator during impact with the surface has

been conducted using computer modelling and laboratory impact tests. Arc-jet tests have been made of the MNL Thermal Protection System (TPS). In addition computer modelling of the heat shield surface and payload temperatures have been performed during the descent. Computer modelling was used to assess the thermal response of the T-IBU during the descent.

The thermal protection system shown in Fig. 12 consists of three sections. Section I covers the

rigid central frontal structure and the inflated torus structure around the central compartment. Section I is constructed of a double layer of KT-11 cloth that can survive temperatures up to 1500 K. The



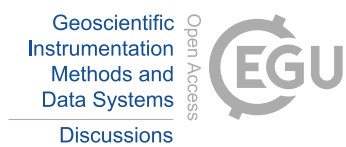

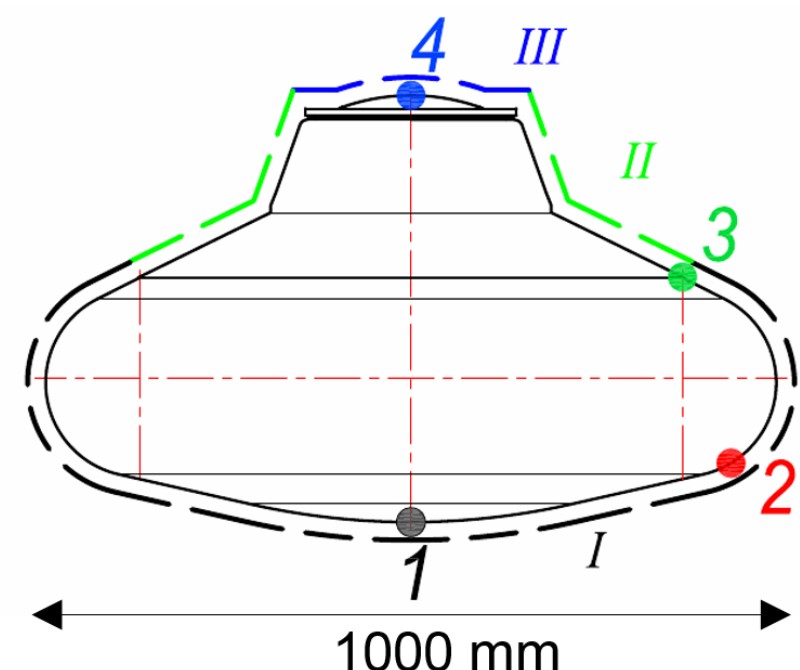

**Fig. 13.** A schematic of MetNet showing the section of the heat protection sections of the inflated aeroshell.

exterior surface of the outer layer of the cloth is covered with a 1-1.3 mm thick layer of material that can absorb the heat experienced during atmospheric entry by sublimating. The sublimation temperature for this material is 950 K. Multi-Layer Insulation (MLI) is situated under the KT-11

cloth layers and is covered with a glass fabric that can survive short duration temperatures of 750-800 K.

Section II consists of MLI covered with a 1 mm thick layer of sublimating material that sublimates at a temperature of 950 K. Section III consists of a rigid lid covered with a 4 mm thick layer of thermal protection material that is radiolucent.

The MNL was found to be statically and dynamically stable under conditions expected during the descent through the atmosphere from wind tunnel and computer modelling. An important system attribute of the lander is its stability during flight as this will influence the drag and lift parameters which will in turn affect the uncertainties in predicting its trajectory through the Martian atmosphere.

The drag coefficient of the heat shield, at zero angle of attack, has been determined experimentally

(Heilimo et al., 2014) and computed to vary from 1.4 down to 1.0 at Mach 4 and 1 respectively. The drag coefficient with the tension cone deployed after the heat shield has been jettisoned has been





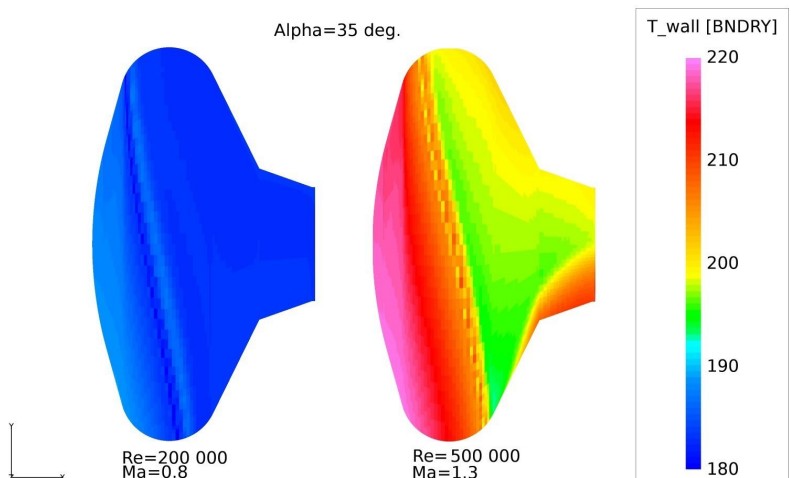

**Fig. 14.** Thermal modelling of the MetNet heat shield.

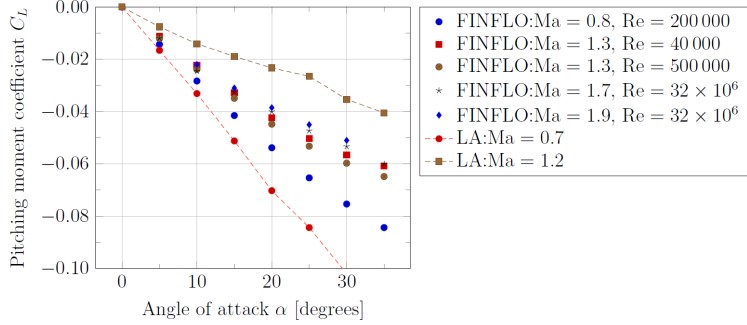

**Fig. 15.** Comparison between the pitching moment coefficients for the heat shield case obtained by numerical simulations using FINFLO numerical code developed by FinFlo Ltd and from LA.

determined to vary from 1.1 to 0.9 at Mach 0.8 and Mach 0.2 respectively. The ballistic coefficient controls the impact speed of the penetrator. With the tension cone deployed the ballistic coefficient is between 19 and 22 kg m$^{-2}$.

Trajectory calculations suggest the minimum entry angle for which the lander will not fly back into space is 5.8°. This is assuming a ballistic coefficient of 20 kg m$^{-2}$, an entry altitude of 120 km and entry speed of 4586 m s$^{-1}$. Minimum and maximum flight angles suitable for a landing that have been investigated are 9.5° and 13.7° respectively. It has also been calculated the flight path angle when the tension cone is deployed will be 60° and at an altitude of 12 km, assuming



the MIBD has been flying through the atmosphere with zero angle of attack. This will allow ample
altitude for the trajectory to turnnover and enabling the penetrator to impact the surface vertically.
The maximum surface temperature of the rigid TPS surface during the steepest trajectory has been
calculated to be 523 K which is well below the short-term tolerance of the rigid TPS. See section
3.1.1 for more details on the TPS.

## 4 Potential mission types

### 4.1 Overview

The MNL design features and characteristics both place constraints and open opportunities for im-
plementation mission types and scientific investigations (Table 3). This section assesses the design
from that point of view and presents some feasible mission types.

As a result of previous flight qualification activities the preparation of a flight-ready spacecraft
(vehicle structure and its descent system) is estimated to take 2-4 years. With an entry mass of about
22.2 kg per unit the MNLs can be easily deployed from a wide range of transfer vehicles. The MNL
structure allows the manufacturing of additional MNL units on short notice and at reasonable cost.
The entry and descent systems could also be used independently from each other in other lander
designs.

Planetary protection requirements are a factor in selection of possible mission types and their
landing sites. Being a Mars *lander* any MNL-based mission is a category IV mission (COSPAR
planetary protection policy). A MNL with a non-biological payload falls as a baseline into sub-
category IVa with the least stringent sterilisation requirements. Since a MNL's landing involves
penetration into the subsurface and depending on the payload complement & the targeted landing
area, a MNL mission may also fall in the more stringent subcategories IVb and IVc.

Even if the payload does not include components aimed at studies of Martian extant life, any
mission the a Martian *special region* (for definition and examples see COSPAR planetary protection
policy) will have to comply with category IVc requirements. The hard landing also increases the
probability of inadvertent exposure of the lander interior with the Martian environment and conse-
quently for any special region mission the entire MNL would have to be sterilised to the Viking
post-sterilisation biological burden. Small size of MNL facilitates comprehensive sterilisation of the
entire lander.

### 4.2 Atmospheric observation networks

A long-duration (possibly an uninterrupted time series) and global coverage *in situ* atmospheric sci-
ence *network* comprising a truly significant number – from 10–20 to several tens – of observation
points on the Martian surface has for a long time been the logical next step for the observational
studies of dynamical features of the Martian atmosphere. Several different network concepts – dif-



**Table 3.** Key MNL design characteristics along with the associated impact (+/-) and mission design constraints.

| Characteristic | Impact | Impact description, constraints |
|---|---|---|
| Low unit mass | + | Increased launch opportunities (piggy-backing), deployment of multiple units with single launch |
| Low unit mass | + | Enables the establishment of an investigation consisting of multiple payloads on the Martian surface |
| Small payload volume | - | Limited set of feasible instruments, limited number of instruments per MNL |
| Entry from interplanetary trajectory or parking orbit | + | Adaptability to different mission concepts and landing accuracy requirements |
| High impact acceleration | - | only robust instrument concepts feasible, modification of the subsurface materials adjacent to the lander surface |
| Stable thermal environment in the payload volume | + | reduced instrument thermal control requirements |
| Stable near-vertical attitude after landing | + | |
| Limited control over the post-landing attitude | - | |
| Access to subsurface layers | + | enables subsurface observations without digging or drilling |
| Limited electrical power and power storage | - | Limits instrument selection and operations |
| Both up- and and downlinks require a relay orbiter | - | Constrains feasible mission concepts |
| Requirements placed on the carrier spacecraft? | | |
| Electromagnetic cleanliness? | | Constrains feasible instruments and their measurement accuracy? |
| Planetary protection | | Constrains types of instruments and allowed landing areas |



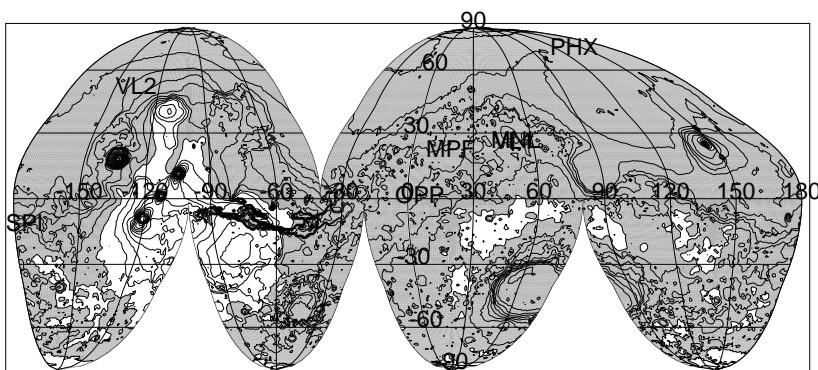

**Fig. 16.** Areas with low enough surface altitudes (¡ 2 km above the datum) suitable as MNL landing sites are shown in grey. Shown are also landing areas of Viking Landers (VL1, VL2), Mars Pathfinder (MPF), Opportunity (OPP), Spirit (SPI), Phoenix (PHX), Mars Science Lander and of a proposed MNL precursor mission.

fering in size and complexity of individual landers as well as number of landers forming the network
– have been proposed (MESUR study report; Chicarro et al., 1993; Masson, 1993; Merrihew et al., 1996; Haberle and Catling, 1996; Harri et al., 1999).

A spatially wide and comprehensive network alone would provide a significant leap in spatial and temporal (from diurnal to seasonal and up to interannual scales) characterisation of the global circulation patterns and the major climatological cycles (dust, $H_2O$ and $CO_2$). The MNL concept
offers a cost-effective and hence realistic element and tool for deploying such a network. The potentially large number of observation points combined with careful selection of locations would permit analyses taking into account also correlations between observations. Such a set of observations has also the potential for providing sufficient constraints to become useful for assimilation into and with Mars Atmospheric Circulation Models.

Optimal locations of observation posts depend on the total number of the network elements. If a network consists of only a few observation posts, it is worthwhile to either create a small local network or to place the posts on different types of terrain and latitudes. This would encompass differences in altitude, latitude and type of surface. In all cases we should place observation posts also on the locations, where observations were previously performed by the Viking Landers (VLs),
the Mars Pathfinder or the Mars Science Laboratory (MSL). This would enable us to compare the current atmosphere at those sites to the atmospheric conditions prevailing earlier. This would be especially interesting at the VL and MSL sites, where long-duration observations are available.

Even though the initial emphasis and focus is from atmospheric science point of view naturally in





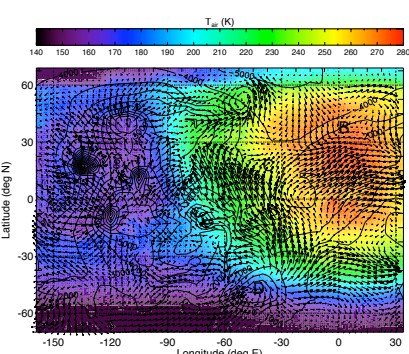

**Fig. 17.** A schematic illustration of a four-lander subnetwork deployed into the equatorial region. The left figure shows the simulated near-surface temperature ($T_{air}$) and wind fields ($V$) along with the labels (A–D) of four possible landing sites. The panels in the right figure show the diurnal variations of (on the left side) $V$ and (on the right side) $T_{air}$ (solid line), ground temperature ($T_g$; dash-dotted line) and surface pressure ($p_s$; dashed line) at the locations A–D and at approximately $L_s =$. All times are shown in "Mars Universal Time" (MTC; local time at Mars' zero meridian). The data shown have been simulated with the Finnish Meteorological Institute/University of Helsinki Mars Limited Area Model (see, *e.g.*, Kauhanen et al., 2008)).

achieving global and long-term coverage, in later stages one can envision deploying clusters of 3–4
landers regionally and with interstation separations of 100–1000 km to form mesoscale *subnetworks*
(Fig. 17). Such subnetworks would be highly useful in more detailed studies of circulation patterns
in regions of particular interest, *e.g.*, the Tharsis volcano area, Valles Marineris, Hellas or perhaps the
circumferences and vicinities of the permanent polar caps. The availability of observations would
allow for regional models to be tuned to the characteristics of that particular region for provisions of
improved understanding of the atmospheric processes and atmosphere-surface interactions, leading
also to more accurate regional forecasts (*see* also section

### 4.3  Joint rover-MNL atmospheric science missions

A subnetwork with interstation dimensions similar to a characteristic range of a longer-range rover
(tens to a hundred km; the network "bracketing" or "interleaving" the area-of-operation of a rover)
offers opportunities for having the MNLs and the rover carrying mutually complementary atmo-
spheric science payloads with intercomparison and intercalibration opportunities. The rover could
for instance carry more complex and resource-hungry instruments (*e.g.*, Mini-TES or LIDAR type)
that the MNLs could not include in their payloads. Such an arrangement would combine in the
studied region stationary longer-term time series measurements with mobile, more advanced and
spatially "sampling" type observations.





### 4.4 Atmospheric observations at high-risk locations

Although correlated and combined observations are qualitatively a leap forward from previous types of observations, the potentially larger numbers of deployable MNLs and acceptance of higher risk of the failure of a single vehicle would permit clear advantages for studies in *microscale* meteo-
rology and atmosphere-surface interactions (*e.g.*, momentum and thermal fluxes): MNLs could be deployed to a large number of very different and also riskier locations and terrains, thus allowing for observations in and of terrains otherwise unlikely to be reachable.

### 4.5 Dedicated ground truth landers for atmospheric sounders

For best accuracy, atmospheric in-orbit sounder observations need so-called ground truth – typi-
cally independent information on for instance the surface pressure. This can be provided by model estimates, but *in situ* observations are most reliable. Hence, having an atmospheric sounder(s) in operation simultaneously with surface observation posts – especially a global surface network – would add considerable scientific value: the combination would provide and combine multipoint surface observations (providing ground truth) with spatially and temporally resolved (4-D) remotely sensed
atmospheric temperature fields. However, the modest resource requirements of MNLs may enable yet another mission approach or emphasis: an advanced atmospheric remote sounder carrying with it a small number of *dedicated* well-placed MNLs for provision of ground truth – perhaps even forming a single mission/launch package.

### 4.6 Other science disciplines

The design offers potential uses in disciplines such as ground studies and seismology. As a MNL lander penetrates the top ground layers, one can envision for instance utilising this in investigations of the top layer of the surface – such as heat fluxes, composition, or depth of permafrost. However, one needs to take into account, that the energies released in the penetration process very likely modify and ground material immediately adjacent to the penetrator.

### 4.7 Pathfinders and precursors for high-value and high-cost missions

Certain Mars mission classes are inherently of extremely high value – i.e., costly in financial terms, of exceptional scientific value or may place humans at risk. Examples include components of sample return missions, In-Situ Resource Utilisation equipment or human missions. Mission assurance is hence of paramount importance and robust ability to observe the weather and atmospheric conditions
are important both for planning and execution of the EDLS phase, the lift-off phase as well as surface operations.

The MNL offers an asset and tool for implementing regional weather observation infrastructure to serve high-value missions. The low cost – especially in comparison with the value of the "prime"





mission – makes deployment of regional weather observation networks composed of MNLs an approach worthy of consideration. Deployment of such a network could conceivably take place during a launch window preceding a high-value mission, thus allowing for collection of a database of weather observations spanning a full Martian year. Such a data set would in turn provide observational basis for development of verified and tuned regional forecast models for the region and provision of high-fidelity forecasts to serve the prime mission components.

## 5   Discussion

**Table 4.** Comparison of MetNet properties and resources to a range of landers, e.g. Ball et al. (2009).

| Lander | Entry | Science | Science/Entry | Size | Entry | impact g | Generate | Store |
|--------|-------|---------|---------------|------|-------|----------|----------|-------|
|        | (kg)  | (kg)    | %             | (cm) | (m s$^{-1}$) | (g) | (Wh) | (Ah) |
| MetNet | 22 | 1.5 | 6.8 | 22 x 22 x 84 | 6.0 | 500 | 14.6 | 40 (Wh) |
| DS2 | 2.73 | 0.15 | 5.5 | 14 x 14 x 22 | 6.9 | 60k | 0 | 0.6 |
| Pathfinder | 584 | 8 + 10.5 | 3.2 | 90 x 100 x 100 | 7.3 | 40 (18.6) | 1200 | 40 |
| Viking | 992 | 91 | 8.2 | 100 x 200 x 50 | 4.5 | 14 | 1680 | 8 |
| MSL | 2401 | 75 | 3.1 | 270 x 300 x 220 | 5.6 | - | 2640 | 84 |

We have developed a Mars lander concept – the MNL – that provides a key landing technology for the future exploration of the environment of Mars. By providing a platform for a 4 kg payload including mechanical structure the MNL is capable of serving various kinds of atmospheric science missions, as well as other kinds of environmental exploration missions

The *MNL* is a semi-hard penetrator utilising inflatable EDLS structures and mechanisms to improve the payload fraction. The mass of the payload bay with its container and thermal insulation is 4 kg with an entry mass of 24 kg. Hence payload fraction of 17 %, which is an excellent number compared to earlier planned Mars landers with similar characteristics [for the Mars-96 penetrators, $F_{pl} < 7\%$ (Surkov and Kremnev, 1998); for the Deep Space 2 $F_{pl}$ appears not to have been reported in open literature]. The design also facilitates thermal control of the payload bay and reduces the number of pyrotechnical devices and commands needed – improving the EDLS reliability.

The semi-hard nature of the entry, descent and landing system provides an excellent payload mass to overall mass ratio of about 0.2 facilitating an efficient use of the mass allocation of a scientific mission. The real strength of the MNL is demonstrated by atmospheric science missions requiring only modest amount of data bandwidth, electrical energy and mass allocation for their scientific payloads. This facilitates the use of a highly versatile payload within the relatively small mass allocation of the MNL vehicle. Furthermore, the MNL EDLS is inherently such that it requires less pyrotechnics (such as explosive bolts) and associated triggering commands than, *e.g.*, a traditional parachute-based landing system. This increases the overall likelihood of mission success.



A major asset of the MNL system is the eventual position, where the payload bay and its outer support structures are penetrated under the Martian surface with only the sensor boom, antenna and the outer rim above the surface. Such a position results in a favorable situation, where the payload bay will be surrounded by a natural thermal environment with temperature ranging from 230 K down to 210 K. These temperatures are still good for the electronics and other parts of the payload with

the exception of batteries that need to be protected with additional thermal system. The position underneath the surface is extremely advantageous for a small probe like MNL from a thermal design point of view. At the Martian surface a small payload with low thermal inertia would require heating systems to survive the low nighttime temperatures of the order of 170–190 K over a wide range of latitudes. The additional heating system would eat up a large fraction of the payload mass. Hence

the MNL concept is giving both thermal shelter and a correct operational position for the payload.

The MNL EDLS allows for deployment to the Martian surface either directly from an interplanetary (hyperbolic) trajectory or from an orbit around Mars. Deployment from orbit enables more accurate landing, whereas direct deployment gives a wider selection of landing sites with the same $\Delta v$ budget. Due to fuel mass savings, direct deployment is often an appealing option – especially

for atmospheric science missions for which modest landing precision is often adequate.

Presently two complete MNL EDLS systems have been manufactured and tested. They will be used on the forthcoming MetNet Precursor missions to demonstrate and validate the robustness and efficiency of the design. Prior to the launches parts of the MNL effected by shelf life, such as the fabrics of the inflatable EDLS components will be replaced or refurbished. The Precursor landers

will also carry out scientific observations and the development of two sets of atmospheric science payloads is currently under way. The payloads and their observations will be described in a separate paper.

The eventual goal of the MetNet Mission concept is to create a network of MNL at the Martian surface operating simultaneously. Eventually a network is needed which the MNL is ideally suited.

MNL could facilitate the mission with perhaps the launch of 15 units during one launch opportunity.

## 5.1   MNL concept validation: precursor missions

The eventual validation of the MetNet Lander vehicle concept calls for an actual mission to the Martian surface and operations at Mars. The first concrete steps in this direction have already been taken. Currently a MMPM with one deployed MNL is being planned. The MMPM would perform contin-

uous scientific observations by using a versatile set of science instruments, but the primary objective of this mission is to demonstrate the feasibility and technical robustness of the MNL concept before building the planet-wide network of observational posts.

For the precursor missions this is extended to include also a 3-axis gyroscope device. Additionally a Solar Incident Sensor with a wide range of dedicated wavelength filters, an optical dust sensor, a

3-axis magnetometer and a radiation monitor are included in the first units payload.


There exist also plans to deploy a network of some tens of MNLs furbished with atmospheric science instrumentation operating simultaneously and focused on the investigations of the Martian atmosphere. This kind of network mission has been planned for several decades (REFs), but since now has never been implemented. The MNL concept provides a suitable tool to achieve this long-standing objective.

The eventual scope of the network mission is to operate the multiple scientific payloads at the Martian surface simultaneously for several Martian years. The MNL provides the means to this objective as it is designed to be operational during several Martian years.

A grand goal of creating a network of observational posts at the Martian surface can be reached either by sending a large number of MetNets to Mars onboard a single mission, or by sending MetNets to Mars in successive launch windows. The latter network generation scheme requires that the lifetime of MetNet vehicles is of the order of several Martian years.

Individual or stand alone MetNet missions can make important scientific investigations characterising the Martian environment.

## 6   Summary and Concluding remarks

Mars Network Lander (MNL), a small semi-hard lander/penetrator design with a payload mass fraction of approximately 17 % has been developed, tested and prototyped. The MNL features an innovative Entry, Descent and Landing System (EDLS) that is based on inflatable structures capable of decelerating the lander from interplanetary transfer trajectories down to a surface impact speed of $50$–$70\,\mathrm{m\,s^{-1}}$ and a deceleration of $< 500\,\mathrm{g}$ for $< 20\,\mathrm{ms}$. The orientation of the penetrator main body into the surface strata at impact is approximately vertical and since the payload bay will be embedded in the surface materials, the bay's temperature excursions will be much less than if it was fully exposed on the Martian surface. The total mass of the prototype design is $\approx 24\,\mathrm{kg}$, with $\approx 4\,\mathrm{kg}$ of mass available for the payload. The MNL is particularly well suited for delivering meteorological and atmospheric instruments to the Martian surface. The possibility exists for sending other environmental instruments. The small size and low mass of an MNL makes it ideally suited for piggy-backing on larger spacecraft. MNLs are designed primarily for use as surface networks but could also be used as pathfinders for high-value landed missions.

*Acknowledgements.*



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
