# Peer review of "The MetNet vehicle: A lander to deploy environmental stations for local and global investigations of Mars"

_Geoscientific Instrumentation, Methods and Data Systems, 2016_

## Referee Comment (RC1) · Anonymous Referee #1 · 16 Aug 2016

This paper deserves to be published once the comments below have been addressed. It promises to give a good overview of a mission concept of interest for flight initially as a demonstrator / piggyback and then as a full network deployment.

Abstract, lines 5-6: I suggest 'simultaneous, distributed in situ measurements' (set of points rather than 2D spatial coverage).

Mass breakdown: mass fraction of approximately 17% - does this include mass maturity margin and payload system margin?

p1 line 24: savings on mass - Potentially, yes - but not only mass; depends on approach to thermal qualification. It might instead just give you a wider range of qualified

components to choose from.

p1 lines 29-31 and p2 lines 114-118: I suggest to cite Ralph Lorenz's relevant paper in these two places: doi:10.1016/j.asr.2011.03.033.

p1 line 60: Seismology being another area (microseismometer).

p1 line 69: 'efficiently' - yes, albeit not precisely. Actual burial depth and thus thermal environment will depend on the surface properties at the impact site.

p2 lines 78-79: Reword and be more precise vs. the impact speeds foreseen for the Mars 96 and DS-2 designs.

p2 lines 82-90: Reword based on the actual order of the sections and sub-sections that follow.

p2 line 95: 'it payload' - change to 'its payload and critical systems'.

p2 lines 114-118: Also mention MarsNet hard lander/penetrator as first European study of such a vehicle? See yellow book, Chicarro, A., Coradini, M., Fulchignoni, M., Liede, I., Lognonne, F., Knudsen, J.M., Scoon, G.E.N., Wanke, H. MARSNET Assessment Study Report, ESA Publication SCI(91)6, European Space Agency, Noordwjik, The Netherlands, January 1991.

[MAJOR] p3 lines 184-185 and p16 lines 877-878, 888-889: Pyros are very reliable components, so surely these are not the major driver of overall reliability (show me an EDL failure attributed to a pyro failing...)... I would expect the reliability to be gauged rather by EDL outcome when parameters of the vehicle's entry state, physical characteristics, GNC approach and atmospheric conditions are dispersed over the expected ranges in a Monte Carlo simulation. In other words, one could have perfect pyro performance (or zero pyros) but still have an unreliable system!

p3 lines 185-186: 'control commands' - What's meant here? EDL on-board control steps (e.g. time based or event-triggered)?

p3 lines 186-189: Again, mass is just one impact of reduction in T range reqt. - see comment above for p1 line 24.

p3 lines 195-196: Relative or inertial entry speeds? Does this limit the interplanetary trajectories that can be used?

p3 line 223: estimates of what?

p4 line 258: 'deployed during the entry phase' - triggered how, e.g. at what g load or Mach no.?

p4 line 260: +18° FPA is a rising, not descending trajectory! Check!

p4 line 262 & p5 line 273: altitude above what level? 0km MOLA or (as perhaps indicated by line 273) pressure-defined?

p5 line 282-283: I assume this 500g load is limited by the piston-like shock-limiting mechanism and its stroke length. What loads are experienced by the external shell?

p5 lines 307-309: Unclear wording... the forebody is stowed, ... once deployed...?

p5 lines 345-348: What is the rigid TPS material used? Only that of the flexible TPS is mentioned.

p6 Figure 3 caption: add legend for the numbered labels.

p7 Figure 4: (b) seems to be covering up something 842mm wide in (a)?

p8 Figure 9 caption: 'The light grey bars...' - Surely the other way round, as per legend? The light grey ranges are smaller.

[MAJOR] p8 Figure 9 caption: The depths shown, are they for the natural, undisturbed surface materials, or do they take into account the thermal short circuit introduced by the presence of the lander (thermally conductive metal structure)? This makes an important difference to the temperature environment the equipment has to withstand.

p8 line 476: REFERENCE missing.

p8 line 493: DEFINition missing.

[MAJOR] p9 lines 512-515: A preliminary power profile would be useful to illustrate the standby, measurement and data relay operations, and demands for heating, e.g. of battery and day vs. night.

p11, line 648: 'temperatures up to 1500 K' - What's the peak heating experienced (W/cm2)?

p11, line 679: 4586 m/s - isn't this too slow for hyperbolic entry? Please check.

[MAJOR] p13 lines 726-728: Does the comprehensive sterilisation of the entire lander include that of the batteries, which presumably have a max. non-op. T below that needed for sterilisation by DHMR? Please clarify bioburden control approach vs. AIT constraints.

p13 lines 739-741: I think the Pascal Mars Scout mission proposal (Haberle et al., ~2000) deserves inclusion in this list.

p16 line 956: REFs missing.

[MAJOR] Please provide an estimate of the data volume that could be relayed. How often could a relay pass be supported, from an energy point of view?

[MAJOR] Please clarify if the MNL is under normal circumstances expected to go dormant waiting for sufficient energy to charge its battery and start operations again, and thus has to wake up with no knowledge of the time. Does the MNL never know ahead of time when a relay pass is expected and thus relies on overlap of its 'link check' status with a relay pass of an orbiter? Given that this is presumably only for a few minutes each sol, doesn't the MNL waste quite some energy listening for a signal? Or is the MNL always expected to keep track of the time and when the next relay passes are?

I also suggest that some of the elements of the MNL and the configuration at each step of the EDL sequence could be clarified by the inclusion of a product tree or block

diagram. This would help understanding section 3.

Some typos for correction:

Abstract, line 2: phenomena, plural

Abstract, line 9: 'number of launches' rather than 'amount of launchers'.

p1 line 20: orient

p2 line 123: mission, singular.

p2, line 125: Mars 96 was never meant to achieve a *stable* Earth orbit, I don't think, only a temporary orbit before the Earth escape burn. Better to say 'failed to achieve Earth escape trajectory'.

p5 line 319: unit, not unity.

p8 Figure 9 caption: modelled, not modellied.

p9 line 518: telescopic, not telescope.

p11, line 644: Fig. 13, not 12, I think.

---

## Referee Comment (RC2) · R. D. Lorenz (Referee) · 7 Sep 2016

This paper is a useful and interesting summary of the MetNet vehicle design. With a few additional details and minor clarifications it should be suitable for publication. line 75 it would be appropriate to cite here Lorenz, R., Planetary Penetrators : Their Origin, History and Future, Advances in Space Research, 48, 403-431, 2011 which gives many pertinent details.

line 85 – since no real definition (g-threshold ?) of semi-hard landers is given, was Mars Pathfinder (which was after all, a Pathfinder for the MESUR mission, which was a network much as intended here) a 'semi-hard lander' ?

Table 1 – should identify where in this classification (if anywhere) the Mars-96 penetrators sit..

line 150 - what is the spin rate required for stability ? How is it effected (spin carrier? spin-eject mechanism? post-separation spin rocket?)

line 164 – seems to be a typo (+18 degrees would be flying away from Mars) – please check the range, and indicate for which altitude the flight path angle is defined .It would be nice here to indicate what the heat fluxes expected at this speed/angle might be line 210, 219 and following. I am not familiar with the alloy designations (Russian, perhaps). Can you indicate the composition or Western equivalents?

Fig 3. – please provide a list of components to go with the numbered labels

line 228 what is TPM-8 fabric (polyimide? polyester?) What is the inflation gas?

Fig 5 – what does label E indicate?

line 304 – please indicate what microcontroller is used, or at least give some specifications (e.g. clock rate, instruction set, memory etc.)

line 311 – please follow your own instructions and define JTAG

line 382 – indicate how accurately Phobos eclipse timings need to be to refine location (e.g. timing to 1 second is a constraint to 1 km??). What kind of on-board clock is used, and how accurate is it (assuming temperature history is known, since most quartz oscillators are strongly temperature-dependent)

406 – what kind of material is KT-11 cloth ?

414 'radiolucent' – not sure if this is a word. 'radio-transparent' is clear.

420 et seq. The shape of the inflatable entry system is novel, so any additional details on aerodynamic coefficients (especially stability derivatives) would be welcome additions to the paper

435 – the planetary protection section is very important, but not very explicit. The considerations given here are worthwhile, but if (as one might hope) the MetNet vehicle and/or some of its components have been qualified for specific planetary protection procedures (e.g. ethylene oxide, DHMR, etc.) that would be well worth stating in detail here – presumably part of the goal of this paper is to indicate mission-readiness !

line 505 - the Lorenz paper mentioned above discusses the 'N of M' survival problem in the context of hard landers and network missions. Some of the considerations there might usefully be raised in this section.

line 545 et seq. The payload mass fraction really needs to be defined better – for small vehicles the level of integration is very high (one reason the DS-2 number is not really given in the literature – the instruments were not boxes that were weighed and bolted on, but integral parts of the vehicle). This confusion is evident from the fact that the numbers in the text ('17%' – including thermal insulation and container) are discrepant with Table 4 immediately above (does this include these other items?) Clarify/check, please.

You may release my name to the authors Ralph Lorenz

---

## Author Comment (AC1) · 18 Nov 2016

Author's esponse to the review (Reviewer #1) of the manuscript

Title: The MetNet vehicle: A lander to deploy environmental stations for local and global investigations of Mars Author(s): A.-M. Harri et al. MS No.: gi-2016-19

Dear Reviewer and the Associate Editor,

Thank you very much for your valuable comments in reviewing this manuscript. We have taken into account your comments and recommendations, and most of them have resulted in modified and/or added text in the manuscript. This response is structured such that we have firstly responded to the major comments and then to minor com-

ments. Our response treats all the reviewer's comments individually by introducing firstly the comment, then our response, and finally, the changes in manuscript are depicted in the end of this response as a supplemented file (in the form of a 'difference mansucript').

Note: text in bold is our response. Text in bold and in quotes is text that can be found in the updated paper (difference manuscript also in the end of this file).

The reviewer comments, not in bold, contains numbers in brackets. These numbers are the page and line numbers of the submitted manuscript that can be found online in the GI discussions section. Reviewer #1 may have received an earlier copy of the manuscript as his page and line numbers do not tie in with the published one in the discussions section. So we have added the correct page and line numbers to help guide the reviewer and ourselves.

Thank you very much for your valuable review effort, A.-M. Harri et al.

Response to reviewer #1 ===============

This paper deserves to be published once the comments below have been addressed. It promises to give a good overview of a mission concept of interest for flight initially as a demonstrator / piggyback and then as a full network deployment.

MAJOR COMMENTS

[MAJOR] p3 lines 184-185 and p16 lines 877-878, 888-889 (P5 116): Pyros are very reliable components, so surely these are not the major driver of overall reliability (show me an EDL failure attributed to a pyro failing...)... I would expect the reliability to be gauged rather by EDL outcome when parameters of the vehicle's entry state, physical characteristics, GNC approach and atmospheric conditions are dispersed over the expected ranges in a Monte Carlo simulation. In other words, one could have perfect pyro performance (or zero pyros) but still have an unreliable system!

Added text on page 5 line 122 (updated online manuscript):

"Our comparative reliability analysis showed that concept B was significantly more re-liable than concept A. This was due to, amongst other things, the lower amount of pyrotechnique devices required by the concept B."

[MAJOR] p8 Figure 9 caption: The depths shown, are they for the natural, undisturbed surface materials, or do they take into account the thermal short circuit introduced by the presence of the lander (thermally conductive metal structure)? This makes an important difference to the temperature environment the equipment has to withstand.

It is now mentioned that the temperatures are for undisturbed material in the paper as follows.

Page 15 in the legend for figure 9:

"The range of temperatures experienced at different latitudes and depths on Mars over the Martian season in material undisturbed by the MNL."

Page 15 at line 304:

"Since the amplitude of the temperature variations tends to decline fairly rapidly with increased depth for undisturbed material (Fig. 9)."

Author comment to the reviewer:

The figure is for the natural, undisturbed surface materials. The figure caption and the text has been updated to make this clear. In reality it is expected that the metal casing will create a short circuit smoothing the temperature profile. Further work will need to be conducted to further assess the thermal environment around the lander. Colder but less variability.

[MAJOR] p9 lines 512-515 (P319 319): A preliminary power profile would be useful to illustrate the standby, measurement and data relay operations, and demands for heating, e.g. of battery and day vs. night.

Text has been added on page 17 and line 355:

"The MNL operations will be defined such that the average energy consumption does not exceed the energy provided by the solar panels. The main energy drain is the transmitter, which is used at such intervals that allow the charging of the battery in-between transmissions. The MNL components allow for such operational cyclograms to be defined."

[MAJOR] p13 lines 726-728 (P23 446): Does the comprehensive sterilisation of the entire lander include that of the batteries, which presumably have a max. non-op. T below that needed for sterilisation by DHMR? Please clarify bioburden control approach vs. AIT constraints.

Text has been added to page 24 line 497:

"The MNL decontamination will be performed via a combination of dry heating and hydrogen peroxide treatment. Dry heating is applied for humidity sensor devices."

[MAJOR] Please provide an estimate of the data volume that could be relayed. How often could a relay pass be supported, from an energy point of view?

Added text at page 16 line 324:

"The battery status monitor together with the system-related part of the software assures that enough energy remains available to perform the essential system tasks like telecommunication link during times of orbiter visibility, and time keeping. Surface to orbit link 16 kbps. The overall data transfer rate is expected to be low; about 0.25 to 0.75 Mb/day on the average, depending on the orbital configuration."

[MAJOR] Please clarify if the MNL is under normal circumstances expected to go dormant waiting for sufficient energy to charge its battery and start operations again, and thus has to wake up with no knowledge of the time. Does the MNL never know ahead of time when a relay pass is expected and thus relies on overlap of its 'link check' status with a relay pass of an orbiter? Given that this is presumably only for a few minutes each sol, doesn't the MNL waste quite some energy listening for a signal? Or is the

MNL always expected to keep track of the time and when the next relay passes are?

Added text at page 16 line 321:

"Operations are designed to make sure the transmitter does not drain the battery. The MNL goes into idle mode to save energy. The clock continues running. At preplanned times the lander waits for a hail signal from the orbiter before transmitting data."

I also suggest that some of the elements of the MNL and the configuration at each step of the EDL sequence could be clarified by the inclusion of a product tree or block diagram. This would help understanding section 3.

Updated figure 2 to more closely follow the configuration at each EDL step.

MINOR COMMENTS

1. Abstract, lines 5-6: I suggest 'simultaneous, distributed in situ measurements' (set of points rather than 2D spatial coverage).

Changed (page 1 line 3)

2. Mass breakdown: mass fraction of approximately 17% - does this include mass maturity margin and payload system margin?

Add text page 29 line 578:

"Hence a payload fraction of 17% based on Engineering Qualification Hardware is an excellent number . . ."

3. p1 line 24 (p1 15): savings on mass - Potentially, yes - but not only mass; depends on approach to thermal qualification. It might instead just give you a wider range of qualified components to choose from.

Updated the text page 1 line 12:

"As the payload bay will be embedded in the surface materials, the bay's temperature excursions will be much less than if it was fully exposed on the Martian surface allowing
a reduction in the amount of thermal insulation and savings on mass."

4. p1 lines 29-31 and p2 lines 114-118: I suggest to cite Ralph Lorenz's relevant paper in these two places: doi:10.1016/j.asr.2011.03.033.

Added "Planetary penetrators: Their origins, history and future" paper reference on page 3 line 82 in the online discussions version of the paper (your p2 lines 114-118):

"A hard-lander, such as high-speed penetrators, typically impact the surface at speeds of around 100 m s-1, and experience high decelerations (1000s of gees) over short time periods during the penetration of the subsurface strata (Lorenz, 2011)."

5. p1 line 60 (p2 38): Seismology being another area (microseismometer).

Agreed, added seismology as an area of investigation. Added this at page 2 line 39:

"Meteorology, climate studies and seismology are areas of investigation that would benefit from a network of observations."

6. p1 line 69 (p2 43): 'efficiently' - yes, albeit not precisely. Actual burial depth and thus thermal environment will depend on the surface properties at the impact site.

Removed the word "efficiently".

7. p2 lines 78-79 (p2 50) : Reword and be more precise vs. the impact speeds foreseen for the Mars 96 and DS-2 designs.

Updated text page 2 line 48:

"This paper describes the MNL concept, a compact and lightweight vehicle designed to deliver a set of instruments to the surface of Mars. The MNL vehicle uses a combination of lightweight inflatable aerodynamic decelerators and a penetrator-like landing system that also give the correct operational attitude. MNL will impact the Martian surface at a relatively lower, and hence safer, speed of around 50 m/s compared to previous high-speed penetrator designs for Mars. For example the Mars 96 and DS2 penetrators had

impact speeds of 80 and 190 m/s respectively, e.g. see Ball et al., (2009a)."

8. p2 lines 82-90 (p2 52): Reword based on the actual order of the sections and sub-sections that follow.

Edited text starting page 3 line 56:

"The paper is organised as follows. In the next section previous Mars landers and their Entry, Descent and Landing System (EDLS) are first reviewed in Section 2.1. MNL development is reviewed in Section 2.2. The selected MNL concept and its EDLS design are discussed and described in Section 2.3. Section 3 provides a more detailed description of the MNL mechanical and electrical systems. Potential mission types and scientific applications of the MNL design are outlined and discussed in Section 4. Future prospects are outlined and recommendations made in Section 5 with precursor missions are outlined in Section 5.1."

9. p2 line 95 (p3 60) : 'it payload' - change to 'its payload and critical systems'.

Changed. See page 3 line 66.

10. p2 lines 114-118 (p3 73): Also mention MarsNet hard lander/penetrator as first European study of such a vehicle? See yellow book, Chicarro, A., Coradini, M., Fulchignoni, M., Liede, I., Lognonne, F., Knudsen, J.M., Scoon, G.E.N., Wanke, H. MARSNET Assessment Study Report, ESA Publication SCI(91)6, European Space Agency, Noordwjik, The Netherlands, January 1991.

Added reference to MarsNet at line 26 page 2.

11. p3 lines 185-186 (p5 115) : 'control commands' - What's meant here? EDL on-board control steps (e.g. time based or event-triggered)?

Edited text (page 4 line 124) and have removed reference to 'control commands'.

"Our comparative reliability analysis showed that concept B was significantly more reliable than concept A. This was due to, amongst other things, the lower amount of

pyrotechnique devices required by the concept B."

12. p3 lines 186-189 (p5 116): Again, mass is just one impact of reduction in T range reqt. - see comment above for p1 line 24.

Added text to page 5 line 126:

"Penetration into the Martian regolith results in the vehicle experiencing reduced diurnal temperature variations. This could help reduce the thermal protection requirement, reducing mass, and in addition permit a wider range of qualified components for use in the vehicle."

13. p3 lines 195-196 (P6 124): Relative or inertial entry speeds? Does this limit the interplanetary trajectories that can be used?

It is the relative entry speed. For higher speeds the MetNet needs to be adjusted slightly to allow for higher entry speeds.nAdded text at page 5 line 132:

"The selected MNL Entry, Descent and Landing System (EDLS) was designed to cope with relative entry speeds of slightly over 6 km/s for the current design configuration. Higher entry speeds are possible with some adjustments to the aerodynamics."

14. p3 line 223 (P6 141): estimates of what?

Mass, text has been updated on page 6 line 151 as below:

"The mass estimates were given with a margin of 10-20%."

15. p4 line 258 (p7 line 163): 'deployed during the entry phase' - triggered how, e.g. at what g load or Mach no.?

Edited text at page 7 line 177 page 7:

"The inflatable heat shield is used during the entry phase to stabilise, decelerate the lander and protect it against excessive heat. The heat shield is inflated using a timer after release from the carrier spacecraft."
16. p4 line 260: +18 FPA is a rising, not descending trajectory! Check!

Thanks, corrected. See page 7 line 177.

17. p4 line 262 & p5 line 273 (P7 165 171): altitude above what level? 0km MOLA or (as perhaps indicated by line 273) pressure-defined?

Yes pressure defined. Updated text. See line 179 page 7.

"The inflatable heat shield diameter is 1 m which decelerates the vehicle down to a Mach number of about 0.85 at an altitude of 4.5-11.0 km above the Martian datum, i.e. the point of zero elevation on Mars equivalent to the altitude where the pressure is 610 Pa, and a dynamic pressure of 95-130~Nm-2 (both altitude and dynamic pressure depending on the angle of entry)."

18. p5 line 282-283 (p7 173): I assume this 500g load is limited by the piston-like shock-limiting mechanism and its stroke length. What loads are experienced by the external shell?

Updated text at page 7 line 187:

"Peak deceleration of the MNL payload bay during the impact will be <500 g, with the outer shell experience about twice the load on the payload, and the total impact time is 20 ms."

19. p5 lines 307-309 (p8 193) : Unclear wording... the forebody is stowed, ... once deployed...?

Updated text page 8 line 208:

"The forebody is stowed inside the surface module cylindrical structure. When the forebody is deployed the empty space provides room for the deceleration of the equipment compartment along a set of crushable rods during the impact with the surface."

20. p5 lines 345-348: What is the rigid TPS material used? Only that of the flexible

TPS is mentioned.

The rigid TPS material, i.e Rigid Aerodynamic Shielding (RAS), can be found in section 3.1.1 Entry and descent related systems.

21. p6 Figure 3 caption: add legend for the numbered labels.

Updated legend:

"A schematic of MetNet showing the section of the heat protection sections of the inflated aeroshell. Section I covers the rigid frontal structure and inflated torus that supports the outer part of the heat shield. Section II covers the rear of the MNL with section III covering the very back."

22. p7 Figure 4: (b) seems to be covering up something 842mm wide in (a)?

Updated figure to remove the hidden image.

23. p8 Figure 9 caption: 'The light grey bars...' - Surely the other way round, as per legend? The light grey ranges are smaller.

Yes corrected.

24. p8 line 476 (P15 300): REFERENCE missing.

Added reference.

25. p8 line 493: (P15 311) DEFINition missing.

Added definition: "JTAG (Joint Test Action Group)"

26. p11, line 648 (P20 406): 'temperatures up to 1500 K' - What's the peak heating experienced (W/cm2)?

Updated text line page 23 line 471 with peak heating rate:

"The maximum surface temperature of the rigid TPS surface during the steepest trajectory has been calculated to be 523 K and a heat flux of 190 kW m-2 which is well

below the short-term tolerance of the rigid TPS."

27. p11, line 679 (P22 427): 4586 m/s - isn't this too slow for hyperbolic entry? Please check.

Yes this is orbital. Updated text at page 23 Line 467:

"This is assuming a ballistic coefficient of 20 kg m-2, an entry altitude of 120 km and an orbital entry speed of 4586 m s-1."

28. p13 lines 739-741 (P23 460): I think the Pascal Mars Scout mission proposal (Haberle et al., 2000) deserves inclusion in this list.

Added reference page 26 line 506.

29. p16 line 956 (P30 598): REFs missing.

Deleted as references are included previously.

Some typos for correction:

Abstract, line 2: phenomena, plural DONE Abstract, line 9: 'number of launches' rather than 'amount of launchers'. DONE p1 line 20: orient DONE p2 line 123 (P3 L77): mission, singular. DONE p2, line 125 (P3 L78): Mars 96 was never meant to achieve a *stable* Earth orbit, I don't think, only a temporary orbit before the Earth escape burn. Better to say 'failed to achieve Earth escape trajectory'. DONE p5 line 319 (P8 L202):: unit, not unity. DONE p8 Figure 9 caption: modelled, not modellied. DONE p9 line 518: telescopic, not telescope. DONE p11, line 644 (P20 L404): Fig. 13, not 12, I think. Yes, DONE.

************************************* *************************************

Modified Manuscript with changes tracked

provided as a supplement file

************************************* *************************************

Interactive
comment

[Figure]

Please also note the supplement to this comment:
http://www.geosci-instrum-method-data-syst-discuss.net/gi-2016-19/gi-2016-19-AC1-supplement.pdf

---

## Author Comment (AC2) · 18 Nov 2016

Author's esponse to the review (Reviewer #2) of the manuscript

Title: The MetNet vehicle: A lander to deploy environmental stations for local and global investigations of Mars Author(s): A.-M. Harri et al. MS No.: gi-2016-19

Dear Reviewer and the Associate Editor,

Thank you very much for your valuable comments in reviewing this manuscript. We have taken into account your comments and recommendations, and most of them have resulted in modified and/or added text in the manuscript. This response is structured such that we have firstly responded to the major comments and then to minor com-

ments. Our response treats all the reviewer's comments individually by introducing firstly the comment, then our response, and finally, the changes in manuscript are depicted in the end of this response as a supplemented file (in the form of a 'difference mansucript').

Thank you very much for your valuable review effort, A.-M. Harri et al.

```
* * *
********************          RESPONSE          ****************************
* * *
```

Note: text in bold is our response. Text in bold and in quotes is text that can be found in the updated paper.

1. This paper is a useful and interesting summary of the MetNet vehicle design. With a few additional details and minor clarifications it should be suitable for publication. line 75 it would be appropriate to cite here Lorenz, R., Planetary Penetrators : Their Origin, History and Future, Advances in Space Research, 48, 403-431, 2011 which gives many pertinent details.

Added reference on page 3 line 82:

"Penetrators for a variety of Solar System destinations have progressed to the concept stage although only two designs have actually been launched (Lorenz, 2011)."

2. line 85 – since no real definition (g-threshold ?) of semi-hard landers is given, was Mars Pathfinder (which was after all, a Pathfinder for the MESUR mission, which was a network much as intended here) a 'semi-hard lander' ?

The Pathfinder reference has been added to the paragraph starting on page 4 line 91. The paragraph has also been rewritten.

"Semi-hard landers are vehicles that impact the surface at speeds, and experience subsequent decelerations, that are between those of a soft lander and a hard lander.

Such landers will experience a moderate deceleration of a few hundreds of gees over the time of some tens of milliseconds. Typically low-mass Martian semi-hard landers have thus far used a combination of heat shield, parachutes and airbags, e.g. see Harri et al. (1999); Linkin et al. (1998), for entry, descent and landing. Heavier semi-hard landers, eg. Golombek et al. (1999), have used additional retrorockets at the end of the descent phase to decelerate down to the required impact speed. Semi-hard landers provide a practical solution for deploying planetary surface payloads that include robust geophysical instruments and are especially suited for deploying lightweight sensor systems needed to perform atmospheric science experiments."

3. Table 1 – should identify where in this classification (if anywhere) the Mars-96 penetrators sits

Mars-96 penetrator EDLS is similar to the selected concept except MetNet has an inflatable rather than rigid heat shield. Added reference for Mars 96 to caption for table 1. See below.

"Table 1. The studied MNL EDLS concept candidates. In each concept the entry and descent phase braking devices are jettisoned to reduce decelerated mass. The concept A1 is as Mars-96 Small Stations (Linkin et al., 1998) and similar to the selected concept as the Mars-96 penetrators (Surkov and Kremnev, 1998) which used a rigid heat shield rather than an inflatable one. The column title 'Entry' refers to the hypersonic and supersonic portion of the flight. 'Descent' refers to the subsonic portion of the flight. A 'tension cone' refers to a type of inflatable decelerator shaped so as to contain tensile stresses, e.g. see Clark et al. (2009) for more information."

4. line 150 - what is the spin rate required for stability? How is it effected (spin carrier? spin-eject mechanism? post-separation spin rocket?)

Spring loaded mechanism that ejects the MetNet and at the same time creates the spin. Rewritten paragraph starting on page 7 line 160 to make this clear:

"Since the MNL itself does not have thrusters for trajectory or attitude changes, the carrier spacecraft may also need to carry out attitude change manoeuvers to release each MNL at the correct angle and at the correct time to reach its intended landing area. Stability is obtained from the aerodynamic properties of the vehicle and spinning of the the MNL. The MNL is ejected and given its spin of one revolution every six seconds by a spring loaded mechanism on the carrier spacecraft."

5. line 164 – seems to be a typo (+18 degrees would be flying away from Mars) – please check the range, and indicate for which altitude the flight path angle is defined .It would be nice here to indicate what the heat fluxes expected at this speed/angle might be.

Fixed, thanks.

6. line 210, 219 and following. I am not familiar with the alloy designations (Russian, perhaps). Can you indicate the composition or Western equivalents?

They are Russian GOST designations. Update text with a reference on page 10 line 226.

"The SAS is made of six metallic (AMg3M Aluminium-Magnesium alloy (GOST, 1977)) hollow tubes."

7. Fig 3. – please provide a list of components to go with the numbered labels.

Provided.

"Fig. 3. The Rigid Aerodynamic Shielding (RAS) includes a blunt front shield plate, a toroidal pressure vessel (which stores the H-IBU inflating gas under pressure) as well as supporting structures for both the Surface Module and the entry and descent systems. The labels are as follows: (1) front shield (FS) with TPC; (2) body of FS; (3) H-IBD filling system; (4) H-IBD; (5) lander body; (6) telescopic cone; (7) telescopic cone drive; (8) T-IBD; (9) shock absorber; (10) cover; (11) instrument container."

8. line 228 what is TPM-8 fabric (polyimide? polyester?) What is the inflation gas?

TPM-8 fabric and inflation gas are classified items specified by the Russian Space Agency.

9. Fig 5 – what does label E indicate?

The label E is not important and has been removed.

10. line 304 – please indicate what microcontroller is used, or at least give some specifications (e.g. clock rate, instruction set, memory etc.)

Added text on page 16 line 330:

"The micro controller type used is a freescale micro controller MC9S12XEP100. The micro controller has 1 MByte Flash PROM for program, 64 kByte RAM for data, 4 kByte EEPROM and 32 kByte D-Flash. External memory used in the MetNet DPU: 2 x 128 Mbit serial flash memory. The same type of micro controller is used for DREAMS aboard ExoMars 2016. "

11. line 311 – please follow your own instructions and define JTAG

Defined: JTAG (Joint Test Action Group)

12. line 382 – indicate how accurately Phobos eclipse timings need to be to refine location (e.g. timing to 1 second is a constraint to 1 km??). What kind of on-board clock is used, and how accurate is it (assuming temperature history is known, since most quartz oscillators are strongly temperature-dependent)

Added text after page 21 line 426:

"The sampling frequency is 1 Hz as a compromise between precision and generated amount of data inside the needed measurement window. This leads to a resolution of approximately 3 km depending somewhat on the latitude of the actual landing side. The clock precision and stability is 5 orders of magnitude better than this, so can be

ignored. The 1 Hz sampling rate is hardcoded in the software outside the cyclogram control system to allow absolute time scheduling."

13. 406 – what kind of material is KT-11 cloth?

KT-11 fabric material that is a glass-cloth-based laminate. Updated text on page 22 line 444.

"Section I is constructed of a double layer of KT-11 cloth, a glass-cloth-based laminate (Shalin, 1995), that can survive temperatures up to 1500 K."

14. 414 'radiolucent' – not sure if this is a word. 'radio-transparent' is clear.

Changed to radio-transparent.

15. 420 et seq. The shape of the inflatable entry system is novel, so any additional details on aerodynamic coefficients (especially stability derivatives) would be welcome additions to the paper.

This is work in progress using numerical simulations to calculate the aerodynamic co-efficients.

16. 435 – the planetary protection section is very important, but not very explicit. The considerations given here are worthwhile, but if (as one might hope) the MetNet vehicle and/or some of its components have been qualified for specific planetary protection procedures (e.g. ethylene oxide, DHMR, etc.) that would be well worth stating in detail here – presumably part of the goal of this paper is to indicate mission-readiness!

Added some details on page 24 line 497.

"The MNL decontamination will most likely be performed via a combination of dry heating and hydrogen peroxide treatment. Dry heating is applied for humidity sensor devices."

17. line 505 - the Lorenz paper mentioned above discusses the 'N of M' survival problem in the context of hard landers and network missions. Some of the considerations there might usefully be raised in this section.

A reference has been added here to the penetrator paper. Page 28 line 543.

"Although correlated and combined observations are qualitatively a leap forward from previous types of observations, the potentially larger numbers of deployable MNLs and acceptance of higher risk of the failure of a single vehicle, e.g. see Harri et al. (2012); Lorenz (2011), would permit clear advantages for studies in microscale meteorology and atmosphere-surface interactions (e.g., momentum and thermal fluxes):"

18. line 545 et seq. The payload mass fraction really needs to be defined better – for small vehicles the level of integration is very high (one reason the DS-2 number is not really given in the literature – the instruments were not boxes that were weighed and bolted on, but integral parts of the vehicle). This confusion is evident from the fact that the numbers in the text ('17%' – including thermal insulation and container) are discrepant with Table 4 immediately above (does this include these other items?) Clarify/check, please.

Added text to explain payload mass fraction page 29 line 596:

"Table 4 compares the MNL to other soft and semi-hard Mars landers and their resources. The science payload fraction is listed here rather than the landed payload fraction. It should be noted that for older spacecraft like the Viking lander the level of integration of the instruments is low, i.e. each instrument may be self-contained rather than sharing resources, with an apparently higher payload ratio for newer spacecraft. Also small landers will tend to have a higher level of integration."

***************************************** *****************************************

Modified Manuscript with changes tracked

provided as a supplement file

\*\*\*\*\*\*\*\*\*\*\*\*\*\*\*\*\*\*\*\*\*\*\*\*\*\*\*\*\*\*\*\*\*\*\*\*\*\*\*\*\*\*\*\* \*\*\*\*\*\*\*\*\*\*\*\*\*\*\*\*\*\*\*\*\*\*\*\*\*\*\*\*\*\*\*\*\*\*\*\*\*\*\*\*\*\*\*\*

Please also note the supplement to this comment:
http://www.geosci-instrum-method-data-syst-discuss.net/gi-2016-19/gi-2016-19-AC2-supplement.pdf

[Figure]

**Supplement:**

Manuscript prepared for Geosci. Instrum. Method. Data Syst.
with version 4.2 of the LaTeX class copernicus.cls.
Date: 24 May 2016

[revised manuscript text omitted]

This paper describes the MNL concept, a compact and lightweight vehicle designed to deliver a
50  set of instruments to the surface of Mars. The MNL vehicle uses a combination of lightweight
inflatable aerodynamic decelerators and a  penetrator-like landing system that
also give the correct operational attitude. MNL will impact the Martian surface at a relatively
lower, and hence safer, speed of around 50 s$^{-1}$ compared to previous high-speed penetrator designs for Mars.  Possible uses of the MNL in Mars exploration along with programmatic and science mission aspects are also discussed.

The paper is organised as follows. In the next section previous Mars landers and their Entry, Descent and Landing System (EDLS)  design are discussed and described in Section  2.3. Section 3 provides a more detailed description of the MNL  mechanical and electrical systems. Potential mission types and scientific applications of the MNL design are outlined and discussed in Section 4. Future prospects are outlined and recommendations made in Section 5  with precursor missions outlined in Section 5.1.

**2 Background**

**2.1 Brief overview of Mars lander technologies**

The survivability of spacecraft during landing will largely depend on the spacecraft being able to absorb the impact energy without damaging  its payload and critical systems. Landers can be divided into three catagories with the division of these catagories being defined by the landing speed which is an indicator of the kinetic energy required to be dissipated by the spacecraft's landing system.

A soft lander typically touches down on the surface at a speed of around one metre per second using a rocket propulsion system that is initiated at subsonic speeds to control and reduce the speed for a soft touchdown. The advantage of using a propulsion system is that manoeuvres like hazard avoidance, and pinpoint landings are possible. Examples of soft Mars landers are the Viking, Phoenix and MSL (Soffen and Snyder, 1976; Soffen, 1976; Guinn et al., 2008; Grotzinger et al., 2012) landers. Soft landing technology is required for large payloads, heavy payloads and payloads with components sensitive to high mechanical loads.

A hard-lander, such as high-speed penetrators, typically impact the surface at speeds of around 100 m s$^{-1}$, and experience high decelerations (1000s of gees) over short time periods during the penetration of the subsurface strata. The use of high-speed penetrators for planetary science were first studied in the USA during the 1970s. The Soviet Union seems to have initiated its studies in the 1980s (Ball et al., 2009, Chapter 19). In Europe the MarsNet mission (Chicarro et al., 1993) was the first study of a penetrator/hard lander system in the early 1990s.

Penetrators for a variety of Solar System destinations have progressed to the concept stage although only two designs have actually been launched (Lorenz, 2011). These are the Russian Mars-96 penetrator (Surkov and Kremnev, 1998) and USA's Deep Space 2 Mars Microprobe (Smrekar

et al., 1999). Each  mission included two penetrators riding piggy back on a carrier space-

90 craft. None of these penetrators were successful: the Mars-96 mission failed to reach  Earth escape trajectory and the DS-2 probes' fate after deployment from the Mars Polar Lander is not known. Hard landers provide a platform to take robust science payloads to a planetary surface with a high mass efficiency. This is because the more gently a vehicle lands the more mass is needed for the EDLS to decelerate the vehicle's velocity before the touchdown on the surface.

95 Semi-hard landers are vehicles that impact the surface at speeds, and experience subsequent decelerations, that are between those of a soft lander and a hard lander.  Such landers will experience a moderate deceleration of a few hundreds of gees over the time of some tens of milliseconds. Typically low-mass Martian semi-hard landers

100 have thus far used a combination of heat shield, parachutes and airbags, e.g. see Harri et al. (1999); Linkin et al. (1998), for entry, descent and landing. Heavier semi-hard landers, eg. Golombek et al. (1999), have used additional retrorockets at the end of the descent phase to decelerate down to the required impact speed. Semi-hard  landers provide a practical solution for

105  deploying planetary surface payloads that include robust geo-physical instruments  and are especially suited for deploying lightweight sensor systems needed to perform atmospheric science experiments.

**2.2 MetNet Lander development history and background**

The work on a semi-hard lander design for the MNL started in August 2000. Five different EDLS

110 concepts (Table 1 and Fig. 1) were initially defined as candidates to be studied. The development of the MNL design was performed over a 7 year period from 2001–2008 by a team comprising of FMI, the LA and the Russian Space Research Institute IKI. The Spanish INTA joined the team in 2008. The MNL development work was funded and led by FMI. The MNL concept and key probe technologies were developed and the critical subsystems were qualified to meet the Martian

115 environmental and functional conditions during the years 2002–2005. Development of the required system instrumentation and prototype science payloads to facilitate testing was carried out in 2004–2008.

In the initial phase of the development five different EDLS concepts were assessed from the viewpoint of finding an optimal solution for deployment of small payloads onto the Martian surface. One

120 concept was a traditional, parachute-based and the remaining four utilized inflatable structures in various ways.

Comparative analysis between the five concepts, underlining and emphasising reliability, payload fraction and complexity of test programme, was carried out. The concepts were catagorised into two catagories. Catagory A contained those landers using airbags for landing and catagory B contained

**Table 1.** The studied MNL EDLS concept candidates. In each concept the entry and descent phase braking devices are jettisoned to reduce decelerated mass. The concept  A1 is as Mars-96 Small Stations (Linkin et al., 1998) and similar to the selected concept as the Mars-96 penetrators (Surkov and Kremnev, 1998) which used a rigid heat shield. The column title 'Entry' refers to the hypersonic and supersonic portion of the flight. 'Descent' refers to the subsonic portion of the flight. A 'tension cone' refers to a type of inflatable decelerator shaped so as to contain tensile stresses, e.g. see Clark et al. (2009) for more information.

| Concept | Entry | Descent | Landing | Station type |
|---|---|---|---|---|
| A1 | rigid shell | parachute | airbags | lander |
| A2 | rigid shell | tension cone | airbags | lander |
| B1 | rigid shell | tension cone | internal shock absorber | penetrator |
| B2 | rigid shell | inflatable torus | same as descent | lander |
| B3 | inflatable | attached ballute | internal shock absorber | penetrator |
| selected | inflatable | tension cone | internal shock absorber | penetrator |

125 those landers using other impact shock attenuation mechanisms for landing. These catagories contained a range of variants as shown in Fig. 1 whose EDLS elements are listed in table 1. Variants A1 and B3 were selected for additional, more detailed study. This study resulted in the formation of lander concepts known as concept A and concept B. Concept A was essential variant A1, based on the Mars 96 small station, which employed a rigid heat shield, parachutes and airbags. Concept B

130 was a new formulation of the EDLS that employed an inflatable heat shield, tension cone and penetrator to deliver the lander to the surface.  Our comparative reliability analysis showed that concept B was significantly more reliable than concept

135 A. This was due to, amongst other things, the lower amount of pyrotechnique devices required by the concept B. Penetration into the Martian regolith  results in the vehicle experiencing reduced diurnal temperature variations . This could help reduce the thermal protection requirement, reducing mass, and in addition permit a wider range of qualified components for use in the vehicle. The current MNL design was chosen as it proved to best

140 satisfy the design goals and criteria.

**2.3 Selected Entry, Descent and Landing System concept**

The selected MNL Entry, Descent and Landing System (EDLS) was designed to cope with relative entry speeds of slightly over 6 $\mathrm{km/s}$  for the current design configuration. Higher entry speeds are possible with some adjustments to the aerodynamics. The major components of the EDLS are

145 the Hypersonic Inflatable Braking Unit (H-IBU), the Transonic Inflatable Braking Unit (T-IBU) and

[Figure]

**Fig. 1.** Landing schemes and designs (from left to right and top-down: A1, A2, B1, B2, B3 as in Table 1) investigated during the course of development of the MNL concept.

penetrator. The H-IBU is an inflatable heat shield designed to resist the heat during hypersonic entry into the atmosphere and decelerate the vehicle down to supersonic speeds. The T-IBU is an inflatable device known as a tension cone and is designed to decelerate the vehicle from supersonic speeds, through the transonic region down to subsonic speeds. Soon after the T-IBU is deployed the H-IBU is jettisoned. Once the H-IBU is jettisoned the forebody of the penetrator is deployed and locked into place ready for impact with the surface.

A MNL can be separated from the carrier spacecraft either directly from a Mars-approaching trajectory or from Martian orbit. Depending on the mission concept, a single carrier spacecraft may carry and deploy a single or several MNL. During the Earth-Mars cruise and possible orbital injection the carrier spacecraft provides each MNL with communications (data link) and power (for instance for health checks every few months, software upgrades, etc.) through the Carrier Spacecraft

Interface and Lander Deployment System (CSI-LDS). The CSI-LDS features may vary depending on the number of MNLs carried, the mission concept and the characteristics of the carrier spacecraft.

A proposed MetNet mission with 16 landers (Harri et al., 2007) was made in 2007 as a study

160  for a European Space Agency (ESA) medium class mission. Each lander was allocated a mass of 20 kg plus 10 kg for the spin/ejection mechanisms. The mass estimates were given with a margin of 10-20%.

The Entry, Descent and Landing (EDL) sequence of activities, shown in Fig. 2, begins with the *separation* phase from a few hours to a few days before actual separation from the carrier spacecraft.

165  The MNL batteries (Section 3.2) are charged to capacity and depending on what has been performed during the preceding health check, final parameter updates to the Command and Data Management System (CDMS; *e.g.*, software, cyclograms – see Sections 3.3 and 3.5) may also be made. Just prior to separation the MNL clock is set and the lander is spun up for stability during the entry into the atmosphere. This process takes $< 10$ min to complete.

170  Since the MNL itself does not have thrusters for trajectory or attitude changes, the carrier spacecraft may also need to carry out attitude change manoeuvers to  release each MNL at the correct angle and at the correct time to reach its intended landing area . Stability is obtained from the aerodynamic properties of the vehicle and spinning of

175  the the MNL. The MNL is ejected and given its spin of one revolution every six seconds by a spring loaded mechanism on the carrier spacecraft.

The behaviour of an MNL during the entire EDL is monitored by a combined 3-axis accelerometer and gyroscope instrument. This diagnostic information is transmitted in packets in near-real time to the relay spacecraft (the carrier or a Mars orbiter, if one is suitably positioned during the EDL) via

180  two dedicated beacon antennas (Section 3.3). The CDMS of the MNL connects the radio system first to the outer beacon antenna, after the inflation of heat shield to a second antenna and after landing and deployment of the instrument mast to the main antenna. The data packets include lander identifiers, hence the monitoring system permits overlap or concurrence of EDL phases of multiple landers.

185  The *entry* phase begins when the MNL senses the first indications of interaction with the atmosphere and ends in the transonic (transition from super- to subsonic) speed regime. The inflatable heat shield is  used during the entry phase to stabilise , decelerate the lander  and protect it against excessive heat. The heat shield is inflated using a timer after release from the carrier spacecraft. The optimal range for the entry angle is -16 to  -18 $\pm 2°$. The inflatable heat shield

190  diameter is 1 m which decelerates the vehicle down to a Mach number of about 0.85 at an altitude of 4.5-11.0 km  above the Martian datum, i.e. the point of zero elevation on Mars equivalent to the altitude where the pressure is 610 Pa, and a dynamic pressure of 95-130 Nm$^{-2}$ (both altitude and dynamic pressure depending on the angle of entry). The tension cone is fully inflated and the heat

[Figure]

**Fig. 2.** MetNet Lander (MNL) Entry and landing sequence and configurations at different parts of the sequence: 1) The complete lander in stowed configuration during cruise and coast phase prior to atmospheric entry; 2)  Hypersonic Inflatable Breaking Unit (H-IBU) deployed for atmospheric entry; 3)  Transonic Inflatable Breaking Unit (T-IBU) deployed,  H-IBU and Rigid Aerodynamic Shielding (RAS) not yet jettisoned; 4) Surface Module (SM) in landing configuration with the forebody deployed.

shield released 10 s later allowing the vehicle to stabilise.

195     The *descent* phase begins when the lander speed is below the transonic regime, the inflatable heat shield is ejected and the tension cone is deployed. The tension cone diameter is 2 m, and is used to decelerate the MNL down to a landing speed of 47-55 m/s, depending on the angle of entry, at the Martian datum . The descent phase ends with the contact of the penetrator tip with the surface.

200 Peak deceleration of the MNL payload bay during the impact will be <500 g, with the outer shell experience about twice the load on the payload, and the total impact time is 20 ms. The minimum impact speed required for an operational landing is 50 m/s with a maximum horizontal wind speed of 20 m/s.

    The *landing* phase begins when the tip of the penetrator touches the surface and ends when the

205 lander has come to rest on and is partially embedded in the top layers of the surface. The deceleration experienced by the payload as the lander penetrates the surface is of the order of 500 g. The structures and mechanisms involved in the final phase landing process, comprising of the Shock Absorbing System (SAS) which is used to reduce the g-levels on the instruments, are described in greater detail in Section 3.1.

**3 Description, operation and testing of the prototype hardware**

**3.1 Structures and mechanisms**

The MNL mechanisms are divided into two categories which are a) the Entry, Descent and related subsystems and b) the Landing and surface operation related subsystems. The Entry and Descent System consists of three subsystems which are 1. Rigid Aerodynamic Shielding (RAS) and supporting structure, 2. Flexible Heat Protection (FHP) and 3. H-IBU (see section 2.3), inflation system and load-bearing elements. The landing and surface operation system consists of three subsystems which are 1. T-IBU and gas generator, 2. Surface Module (SM) with a Shock-Absorbing System (SAS) and 3. Equipment Compartment (EC).

During cruise, entry and most of the descent phase, the EDLS related subsystems and the SM are efficiently packed in terms of volume. This is achieved by stowing the systems telescopically inside each others where possible. The forebody is stowed inside the surface module cylindrical structure using the empty volume, once deployed . When the forebody is deployed the empty space provides room for the deceleration of the equipment compartment along a set of crushable rods during the impact with the surface. The forebody will be deployed into a landing configuration after jettisoning the H-IBU. The stowed surface module and forebody are both stowed inside the mechanical support cylindrical structure of the rigid section of the front shield during entry and upper atmosphere braking phase. Fig. 3 shows the complete lander with empty stowed H-IBU wrapped around it.

The SM accommodates the system electronics and payload instruments. The T-IBU is connected to and surrounds the surface module. These three subsystems stay interconnected after touchdown forming the surface operating unity unit. The power system solar cells are attached to the upper surface of the T-IBU. Other subsystems, forming most of the EDLS, are ejected during the descent phase as shown in Fig. 2. The SM accommodates the EC, which houses the system and payload electronics and supports external sensors (the boom with meteorological sensors, optical sensor) and telecoms antenna. The SM includes a rear cover lid, which protects the module during entry and landing.

The SM includes the shock-absorbing system (SAS). This system allows the Equipment Module (EM) to slide some tens of centimetres during the impact with the surface and thus reduce the deceleration experienced by the equipment module by a factor of around two compared to other rigid mechanics such as the surface module body structures. The SAS is made of six metallic (AMg3M Aluminium-Magnesium alloy (GOST, 1977)) hollow tubes. The equipment module slides along these tubes during the impact on the surface and kinetic energy is reduced by squeezing the hollow tubes flat by squared sliding slots of the equipment module supporting adapter.

[Figure]

**Fig. 3.** The Rigid Aerodynamic Shielding (RAS) includes a blunt front shield plate, a toroidal pressure vessel (which stores the H-IBU inflating gas under pressure) as well as supporting structures for both the Surface Module and the entry and descent systems. The labels are as follows: (1) front shield (FS) with TPC; (2) body of FS; (3) H-IBD filling system; (4) H-IBD; (5) lander body; (6) telescopic cone; (7) telescopic cone drive; (8) T-IBD; (9) shock absorber; (10) cover; (11) instrument container.

[revised manuscript text omitted]

 Operations are designed to make sure the transmitter does not drain the battery. The MNL goes into idle mode to save enrgy. The clock continues running. At preplanned times the lander waits for a hail signal from the orbiter before transmitting data.

The battery status monitor together with the system-related part of the software assures that enough energy remains available to perform the essential system tasks like telecommunication link during times of orbiter visibility, and time keeping. Surface to orbit link is around 16 kbps. The overall data transfer rate is expected to be low; about 0.25 to 0.75 Mb/day on the average, depending on the orbital configuration.

The CDMS is built around a fully redundant micro controller system where one system is capable of autonomously detecting and correcting errors in the performance of the active controller. The

[Figure]

**Fig. 10.** The MNL electronics consists of a hot-redundant microcontroller with duplicated interfaces, a power conditioning and a radio system. Communication with the detectors is via serial links.

micro controller type used is a freescale micro controller MC9S12XEP100. The micro controller has 1 MByte Flash PROM for program, 64 kByte RAM for data, 4 kByte EEPROM and 32 kByte D-Flash. External memory used in the MetNet DPU: 2 x 128 Mbit serial flash memory. The same type of micro controller is used for DREAMS aboard ExoMars 2016.

All hardware interfaces and memories are duplicated so that the secondary controller can operate the system completely in case the primary one malfunctions without correction possibility. A block diagram is shown in Fig. 10. The software and even the controller hardware configuration can be updated from the operational controller via the implemented JTAG (DEFINEJoint Test Action Group) input, using the own configuration as reference. The monitoring between the redundant controllers is done by a bi-directional CAN-bus interface integrated into each of the controllers. This link is also used to update cyclogram contents in the secondary controller after a commanded update.

Each of the MNL sensors are required to pre-process its observational data including image compression inside the panorama camera. The controller itself does not include any general data compression software to minimize especially energy resources.

**3.4 Payload resources and strawman payload**

With a total of 4 kg for the payload including 2.6 kg for the control system and meteorological mast including antennas, 1.4 kg are available for the instruments. As instruments are normally not operated in parallel the maximum available current from the batteries has only to be shared between

365 the CPU and one sensor. With a maximum current of 5.8 A at 2 * 3.5V about 40 W maximum power is available for a short time (of the order of 1 h; see also Section 3.2). The average power used has to be balanced against the average power provided by the solar cells and limits the time an instrument can be operated per sol. Instruments and their electronics can be accommodated either inside the thermally stabilized payload compartment guaranteeing temperatures above -50 C or outside on or

370 close to the  telescopic mast. The payload compartment is illustrated in Fig. 11.

The MNL operations will be defined such that the average energy consumption does not exceed the energy provided by the solar panels. The main energy drain is the transmitter, which is used at such intervals that allow the charging of the battery in-between transmissions. The MNL components allow for such operational cyclograms to be defined.

[revised manuscript text omitted]

595 provision of high-fidelity forecasts to serve the prime mission components.

**5 Discussion**

We have developed a Mars lander concept – the MNL – that provides a key landing technology for the future exploration of the environment of Mars. By providing a platform for a 4 kg payload including mechanical structure the MNL is capable of serving various kinds of atmospheric science
600    missions, as well as other kinds of environmental exploration missions

The *MNL* is a semi-hard penetrator utilising inflatable EDLS structures and mechanisms to improve the landed payload fraction which is the payload fraction that could conceivably be replaced. For example if a different mission is envisaged using the MNL. The mass of the payload bay with its container and thermal insulation is 4 kg with an entry mass of 24 kg. Hence a payload fraction of 17
605    % based on Engineering Qualification Hardware is an excellent number compared to earlier planned Mars landers with similar characteristics (for the Mars-96 penetrators, $F_{pl} < 7\%$ (Surkov and Kremnev, 1998); for the Deep Space 2 $F_{pl}$ appears not to have been reported in open literature). The design also facilitates thermal control of the payload bay and reduces the number of pyrotechnical devices and commands needed – improving the EDLS reliability.
610    Table 4 compares the MNL to other soft and semi-hard Mars landers and their resources. The science payload fraction is listed here rather than the landed payload fraction. It should be noted that for older spacecraft like the Viking lander the level of integration of the instruments is low, i.e. each instrument may be self-contained rather than sharing resources, with an apparently higher payload ratio for newer spacecraft. Also small landers will tend to have a higher level of integration. The
615    MNL compares favourable in this respect to other types of landers. The small dimesions of the MNL make it small enough to include multiple units in a carrier spacecraft.

**Table 4.** Comparison of MetNet properties and resources to a range of landers, e.g. Ball et al. (2009). In column 4 the science payload fraction is the science payload mass divided by the entry mass. The letters W, L and H in column 5 stand for width, length and height respectively.

| Lander  | Entry mass (kg) | Science payload mass (kg) | Science  payload fraction %) |  Dimensions W x L x H (cm x cm x cm)  | Entry speed (m s⁻¹) | impact  decel- eration (g) | Generate (Wh) | Store (
[revised manuscript text omitted]

Barderas, G. and Romero, P.: On the inverse problem of determining Mars lander coordinates using Phobos eclipse observations, Planetary and Space Science, 79, 39–44, 2013.

CCSDS 211.0-B-4: Proximity-1 Space Link Protocol – Data Link Layer, Consultative Committee for Space Data Systems, CCSDS Secretariat, Office of Space Communication (Code M-3), National Aeronautics and Space Administration, Washington, DC 20546, USA, 4 edn., 2006.

Chicarro, A. F., Coradini, M., Fulchignoni, M., Hiller, K., Knudsen, J. M., Liede, I., Lindberg, C., Lognonné, P., Pellinen, R., Spohn, T., Scoon, G. E. N., Taylor, F. W., and Wänke, H.: MARSNET Phase-A Study Report, Tech. Rep. SCI(93)2, European Space Agency, 1993.

Clark, I. G., Hutchings, A. L., Tanner, C. L., and Braun, R. D.: Supersonic Inflatable Aerodynamic Decelerators for Use on Future Robotic Missions to Mars, Journal of Spacecraft and Rockets, 46, 340–352, 2009.

COSPAR planetary protection policy: COSPAR planetary protection policy, Policy document, Committee on Space Research (COSPAR), original dated 20 Oct 2002, amended 24 Mar 2005, 2005.

Golombek, M. P., Bridges, N. T., Moore, H. J., Murchie, S. L., Murphy, J. R., Parker, T. J., Rieder, R., Rivellini, T. P., Schofield, J. T., Seiff, A., Singer, R. B., Smith, P. H., Soderblom, L. A., Spencer, D. A., Stoker, C. R., Sullivan, R., Thomas, N., Thurman, S. W., Tomasko, M. G., Vaughan, R. M., Wänke, H., Ward, A. W., and Wilson, G. R.: Overview of the Mars Pathfinder Mission: Launch through landing, surface operations, data sets, and science results, Journal Geophysical Research, 104, 8523–8554, 1999.

Gómez-Elvira, J., Armiens, C., Carrasco, I., Genzer, M., Gómez, F., Haberle, R., Hamilton, V. E., Harri, A.-M., Kahanpää, H., Kemppinen, O., Lepinette, A., Martín Soler, J., Martín-Torres, J., Martínez-Frías, J., Mischna, M., Mora, L., Navarro, S., Newman, C., Pablo, M. A., Peinado, V., Polkko, J., Rafkin, S. C. R., Ramos, M., Rennó, N. O., Richardson, M., Rodríguez-Manfredi, J. A., Romeral Planelló, J. J., Sebastián, E., Torre Juárez, M., Torres, J., Urquí, R., Vasavada, A. R., Verdasca, J., and Zorzano, M.-P.: Curiosity's rover environmental monitoring station: Overview of the first 100 sols, Journal of Geophysical Research (Planets), 119, 1680–1688, doi:10.1002/2013JE004576, 2014.

GOST: Specifications, Tech. rep., 1977.

Grotzinger, J. P., Crisp, J., Vasavada, A. R., Anderson, R. C., Baker, C. J., Barry, R., Blake, D. F., Conrad, P., Edgett, K. S., Ferdowski, B., Gellert, R., Gilbert, J. B., Golombek, M., Gómez-Elvira, J., Hassler, D. M., Jandura, L., Litvak, M., Mahaffy, P., Maki, J., Meyer, M., Malin, M. C., Mitrofanov, I., Simmonds, J. J., Vaniman, D., Welch, R. V., and Wiens, R. C.: Mars Science Laboratory Mission and Science Investigation, Space Sci. Rev., p. 61, doi:10.1007/s11214-012-9892-2, 2012.

Guinn, J. R., Garcia, M. D., and Talley, K.: Mission design of the Phoenix Mars Scout mission, J. Geophys. Res., 113, E00A26, doi:10.1029/2007JE003038, http://dx.doi.org/10.1029/2007JE003038, 2008.

Haberle, R. M. and Catling, D. C.: A Micro-Meteorological mission for global network science on Mars: rationale and measurement requirements, Planet. Space Sci., 44, 13611383, doi:10.1016/S0032-0633(96)00056-6, 1996.

Harri, A., Linkin, V., Polkko, J., Marov, M., Pommereau, J., Lipatov, A., Siili, T., Manuilov, K., Lebedev, V., Lehto, A., Pellinen, R., Pirjola, R., Carpentier, T., Malique, C., Makarov, V., Khloustova, L., Esposito, L.,

Maki, J., Lawrence, G., and Lystsev, V.: Meteorological observations on Martian surface: met-packages of Mars-96 Small Stations and Penetrators, Planet. Space Sci., 46, 779–793, doi:10.1016/S0032-0633(98)00012-9, http://dx.doi.org/10.1016/S0032-0633(98)00012-9, 1998.

Harri, A.-M., Marsal, O., Lognonné, P., Leppelmeier, G. W., Spohn, T., Glassmeier, K.-H., Angrilli, F., Banerdt,
745  W. B., Barriot, J. P., Bertaux, J.-L., Bérthelier, J. J., Calcutt, S., Cerisier, J. C., Crisp, D., Déhant, V., Giardini, D., Jaumann, R., Langevin, Y., Menvielle, M., Mussmann, G., Pommereau, J. P., di Pippo, S., Guerrier, D., Kumpulainen, K., Larsen, S., Mocquet, A., Polkko, J., Runavot, J., Schumacher, W., Siili, T., Simola, J., Tillman, J. E., and the NetLander Team: Network science landers for Mars, Adv. Space Res., 23, 19151924, doi:10.1016/S0273-1177(99)00279-3, 1999.

750  Harri, A.-M., Leinonen, J., Merikallio, S., Paton, M., Haukka, H., and Polkko, J.: MetNet - In situ observational Network and Orbital platform to investigate the Martian environment, Tech. rep., 2007.

Harri, A.-M., Schmidt, W., Romero, P., Vazquez, L., Barderas, G., Kemppinen, O., A. C., Vazquez-Poletti, J. L., Llorente, I. M., Haukka, H., and Paton, M.: Phobos eclipse detection on Mars: Theory and practice, Tech. rep., 2012.

755  Harri, A.-M., Genzer, M., Kemppinen, O., Gomez-Elvira, J., Haberle, R., Polkko, J., Savijärvi, H., Rennó, N., Rodriguez-Manfredi, J. A., Schmidt, W., Richardson, M., Siili, T., Paton, M., Torre-Juarez, M. D. L., Mäkinen, T., Newman, C., Rafkin, S., Mischna, M., Merikallio, S., Haukka, H., Martin-Torres, J., Komu, M., Zorzano, M.-P., Peinado, V., Vazquez, L., and Urqui, R.: Mars Science Laboratory relative humidity observations: Initial results, Journal of Geophysical Research (Planets), 119, 2132–2147, doi:
760  10.1002/2013JE004514, 2014a.

Harri, A.-M., Genzer, M., Kemppinen, O., Kahanpää, H., Gomez-Elvira, J., Rodriguez-Manfredi, J. A., Haberle, R., Polkko, J., Schmidt, W., Savijärvi, H., Kauhanen, J., Atlaskin, E., Richardson, M., Siili, T., Paton, M., Torre Juarez, M., Newman, C., Rafkin, S., Lemmon, M. T., Mischna, M., Merikallio, S., Haukka, H., Martin-Torres, J., Zorzano, M.-P., Peinado, V., Urqui, R., Lapinette, A., Scodary, A., Mäkinen, T., Vazquez,
765  L., Rennó, N., and REMS/MSL Science Team: Pressure observations by the Curiosity rover: Initial results, Journal of Geophysical Research (Planets), 119, 82–92, doi:10.1002/2013JE004423, 2014b.

Heilimo, J., Harri, A.-M., Aleksashkin, S., Koryanov, V., Arruego, I., Schmidt, W., Haukka, H., Finchenko, V., Martynov, M., Ostresko, B., Ponomarenko, A., Kazakovtsev, V., Martin, S., and Siili, T.: RITD - Adapting Mars Entry, Descent and Landing System for Earth, in: EGU General Assembly Conference Abstracts,
770  vol. 16 of *EGU General Assembly Conference Abstracts*, 2014.

Kauhanen, J., Siili, T., Jrvenoja, S., and Savijrvi, H.: The *Mars Limited Area Model* (MLAM) and simulations of atmospheric circulations for the Phoenix landing area and season-of-operation, J. Geophys. Res., 113, E00A14, doi:10.1029/2007JE003011, 2008.

Linkin, V., Harri, A.-M., Lipatov, A., Belostotskaja, K., Derbunovich, B., Ekonomov, A., Khloustova, L.,
775  Kremnev, R., Makarov, V., Martinov, B., Nenarokov, D., Prostov, M., Pustovalov, A., Shustko, G., Järvinen, I., Kivilinna, H., Korpela, S., Kumpulainen, K., Lehto, A., Pellinen, R., Pirjola, R., Riihelä, P., Salminen, A., Schmidt, W., Siili, T., Blamont, J., Carpentier, T., Debus, A., Hua, C. T., Karczewski, J.-F., Laplace, H., Levacher, P., Lognonné, P., Malique, C., Menvielle, M., Mouli, G., Pommereau, J.-P., Quotb, K., Runavot, J., Vienne, D., Grunthaner, F., Kuhnke, F., Mussmann, G., Rieder, R., Wänke, H., Economou,
780  T., Herring, M., Lane, A., and McKay, C. P.: A sophisticated lander for scientific exploration of Mars:

scientific objectives and implementation of the Mars-96 Small Station, Planet. Space Sci., 46, 717737, doi: 10.1016/S0032-0633(98)00008-7, 1998.

Lorenz, R. D.: Planetary penetrators: Their origins, history and future, Advances in Space Research, 48, 403–431, 2011.

785   Masson, P.: InterMARSNET - a contribution to an international network of stations on the surface of Mars, proposal in response to the ESA call for ideas for the M3 mission, 1993.

Merrihew, S. C., Haberle, R. M., and Lemke, L. G.: A Micro-Meteorological mission for global network science on Mars: a conceptual design, Planet. Space Sci., 44, 13851393, doi:10.1016/S0032-0633(96)00055-4, 1996.

790   MESUR study report: Mars Environmental Survey (MESUR) science objectives and mission description., NASA Ames Research Center study report, 1991.

Paton, M. D., Harri, A.-M., Savijärvi, H., Mäkinen, T., Hagermann, A., Kemppinen, O., and Johnston, A.: Thermal and microstructural properties of fine-grained material at the Viking Lander 1 site, Icarus, 271, 360–374, doi:10.1016/j.icarus.2016.02.012, 2016.

795   Romero, P., Barderas, G., Vazquez-Poletti, J. L., and Llorente, I. M.: Spatial chronogram to detect Phobos eclipses on Mars with the MetNet Precursor Lander, Planetary and Space Science, 59, 1542–1550, 2011.

Savijärvi, H., Crisp, D., and Harri, A.-M.: Effects of CO2 and dust on present-day solar radiation and climate on Mars, Quarterly Journal of the Royal Meteorological Society, 131, 2907–2922, doi:10.1256/qj.04.09, 2005.

Savijärvi, H., Harri, A.-M., and Kemppinen, O.: The diurnal water cycle at Curiosity: Role of exchange with
800   the regolith, Icarus, 265, 63–69, doi:10.1016/j.icarus.2015.10.008, 2016.

Shalin, R. E.: Polymer Matrix Composites, Tech. rep., 1995.

Smith, P. H., Tamppari, L., Arvidson, R. E., Bass, D., Blaney, D., Boynton, W., Carswell, A., Catling, D., Clark, B., Duck, T., DeJong, E., Fisher, D., Goetz, W., Gunnlaugsson, P., Hecht, M., Hipkin, V., Hoffman, J., Hviid, S., Keller, H., Kounaves, S., Lange, C. F., Lemmon, M., Madsen, M., Malin, M., Markiewicz, W., Marshall,
805   J., McKay, C., Mellon, M., Michelangeli, D., Ming, D., Morris, R., Renno, N., Pike, W. T., Staufer, U., Stoker, C., Taylor, P., Whiteway, J., Young, S., and Zent, A.: Introduction to special section on the Phoenix Mission: Landing Site Characterization Experiments, Mission Overviews, and Expected Science, Journal of Geophysical Research (Planets), 113, E00A18, 2008.

Smrekar, S., Catling, D., Lorenz, R., Magalhães, J., Moersch, J., Morgan, P., Murray, B., Presley, M., Yen, A.,
810   Zent, A., and Blaney, D.: Deep Space 2: the Mars microprobe mission, J. Geophys. Res., 104, 27 01327 030, doi:10.1029/1999JE001073, 1999.

Soffen, G. A.: Scientific results of the Viking missions, Science, 194, 1274–1276, 1976.

Soffen, G. A.: Scientific Results of the Viking Missions, Science, 194, 1274–1276, doi:10.1126/science.194.4271.1274, http://www.sciencemag.org/content/194/4271/1274.abstract, 1976.

815   Soffen, G. A. and Snyder, C. W.: The First Viking Mission to Mars, Science, 193, 759–766, doi:10.1126/science.193.4255.759, http://www.sciencemag.org/content/193/4255/759.short, 1976.

Surkov, Y. A. and Kremnev, R. S.: Mars-96 mission: Mars exploration with the use of penetrators, Planet. Space Sci., 46, 1689 – 1696, doi:10.1016/S0032-0633(98)00071-3, 1998.

Taylor, P. A., Catling, D. C., Daly, M., Dickinson, C. S., Gunnlaugsson, H. P., Harri, A.-M., and Lange,
820   C. F.: Temperature, pressure, and wind instrumentation in the Phoenix meteorological package, Journal of

Geophysical Research (Planets), 113, E00A10, doi:10.1029/2007JE003015, 2008.